# StratoBayes: A Bayesian method for automated stratigraphic correlation and age modelling

Kilian Eichenseer<sup>1</sup>, Matthias Sinnesael<sup>1,2</sup>, Martin R. Smith<sup>1</sup> and Andrew R. Millard<sup>3</sup>

—

<sup>1</sup>Department of Earth Sciences, Durham University, South Road, DH1 3LE, Durham, United Kingdom

<sup>2</sup>Geology, School of Natural Sciences, Trinity College Dublin, Dublin 2, Ireland

<sup>3</sup>Department of Archaeology, Durham University, South Road, DH1 3LE, Durham, United Kingdom

—

Corresponding author: Kilian Eichenseer - kilian.eichenseer@gmail.com

## **Abstract**

5

7

Stratigraphic correlation and age modelling are fundamental to reconstructing Earth's history, biological evolution, and palaeoclimate, and underpin the exploration for subsurface resources. Correlations are produced by integrating diverse stratigraphic data across multiple sites, typically by visual inspection. Here, we introduce 'StratoBayes', a Bayesian statistical framework that combines stratigraphic correlation and depositional age estimation of stratigraphic horizons, i.e. age modelling. Our method aligns quantitative signals from two or more sites by shifting and scaling, allowing for sedimentation rate changes between stratigraphic partitions. The likelihood of an alignment is evaluated by how well the adjusted signals conform to a shared smooth trend, represented by a cubic spline. Tie points or independent age constraints, such as radiometric dates or biostratigraphic markers, can be integrated within this framework, providing age estimates for all sites. Our approach identifies multiple alignments where distinct alternatives exist, estimates their relative probabilities, and quantifies the uncertainty associated with correlations and age estimates. We apply StratoBayes to a lower Cambrian dataset comprising a combination of  $\delta^{13}$ C records, radiometric dates and astrochronology from four sites in Morocco and Siberia. The results demonstrate its capacity to quantify existing alignments, and provide the first precise age estimate for the evolutionary appearance of trilobites in Siberia, one of the hallmarks of the Cambrian Explosion. Beyond this application, StratoBayes offers a generalisable framework for probabilistic stratigraphic correlation, with potential to improve age models across a range of proxy records and time intervals.

## 1 Introduction

29

Stratigraphic correlation works on the basis that rocks that were deposited under similar conditions or at 31 the same time tend to share characteristics that allow for their attribution to a stratigraphic or temporal 32 horizon. For example, insofar as temporal changes in the global  $\delta^{13}$ C composition of seawater are reflected in marine sedimentary rocks, matching trends of changing  $\delta^{13}$ C in rock sections from different locations 33 34 can be used to place those sections on a relative time scale (Cramer and Jarvis, 2020; Saltzman et al., 2012). 35 Quantitative signals such as isotopic compositions, elemental concentrations or geophysical well-log data 36 present a particular challenge: in aligning those signals by eye, the stratigrapher has to make a large number 37 of intuitive decisions about which peaks and troughs are likely to line up. Trying to integrate all the 38 stratigraphic evidence from multiple sites often results in more than one potential alignment solution and 39 differing interpretations between different workers (Bowyer et al., 2022, 2023; Landing and Kruse, 2017; 40 Smith et al., 2016). 41 Computer algorithms have been designed to address the problems arising from visual correlation 42 (Agterberg, 1990; Lisiecki and Lisiecki, 2002; Rudman and Lankston, 1973). Algorithms designed for aligning quantitative signals from two or more sites typically use a point-based approach, aligning each 43 44 point of site A with zero, one or multiple points from site B. This approach proposes variable sedimentation 45 rates between points. This flexibility in principle allows the most precise alignments, though potentially at 46 the cost of overfitting. Point-based algorithms commonly use dynamic time warping (DTW), a technique 47 that finds the optimal match between two time-series data by adjusting their alignment (Sakoe and Chiba, 1978). For a selection of recent approaches using dynamic time warping for stratigraphic alignment, see 48 49 Wheeler and Hale (2014); Hay et al. (2019); Baville et al. (2022); Sylvester (2023); and Hagen et al. (2024). 50 The limitations of DTW-based approaches are that they commonly require known section tops and bottoms 51 (Sylvester, 2023); and they are generally deterministic, providing only a single solution without any 52 indication of uncertainty or alternative alignments (but see Al Ibrahim, 2022; Hay et al., 2019). The 53 integration of additional stratigraphic information besides the quantitative signals tends to be difficult, requiring extra steps outside of the core DTW-algorithm (e.g. Hagen and Creveling, 2024). 54 55 Probabilistic approaches overcome some of these limitations by estimating the probabilities of different 56 outcomes, rather than producing deterministic predictions. An effective probabilistic approach is offered 57 by the Bayesian framework, which integrates multiple sources of uncertainty by combining prior knowledge, encapsulated mathematically as a prior probability distribution, with a custom likelihood 58 59 function that is used to evaluate the likelihood of observed data. Given an appropriate prior and likelihood

function it is straightforward to integrate different types of stratigraphic information. Bayesian approaches are commonly employed in age-depth models that interpolate between absolute age constraints or tie points; examples include Bchron (Haslett and Parnell, 2008) and Oxcal (Ramsey, 1995). This approach can be extended by incorporating prior expectations on hiatuses, sedimentation rates, and rate variability, including external information such as astrochronological data (e.g. Blaauw and Christen, 2011; Trayler et al., 2024). Recent Bayesian methods have attempted to combine stratigraphic correlation and age modelling. Lee et al. (2022) have implemented a Bayesian method that uses Gaussian process regression to match Cenozoic oxygen isotope data from one site to an oxygen isotope stack, while simultaneously integrating age estimates from radiocarbon dates to produce probabilistic age-depth models (i.e. the BIGMACS model). This method improves upon earlier approaches by specifying uncertainty for tie points and integrates prior knowledge on Cenozoic sedimentation rates with absolute age information from the aligned site. However, age uncertainties from the reference site are not included, and varying sampling resolution or large sedimentation rate changes may violate model assumptions and impede the broader adoption of this method in its current form (Middleton et al., 2024). Edmonsond and Dyer (2024) have developed a different Bayesian method based on Gaussian process regression that works without prior knowledge of sedimentation rates, but requires minimum and maximum age estimates for all sections, and the absence of an explicit prior on sedimentation rates may risk overfitting. Here, we introduce a versatile Bayesian method for stratigraphic correlation and age modelling that can align quantitative signals from two or more sites without the need to specify tie points or top and bottom ages, and with no restrictions on sampling frequencies. Possible sedimentation rates can be specified by the user as priors, and the likelihood encompasses the alignment of the signals and, optionally, additional age constraints such as dated horizons. The method requires only vague prior knowledge on the ages and on the degree of overlap of the sections, along with order-of-magnitude estimates of sedimentation rates; it is not necessary to specify matching section tops or bottoms. The model is able to integrate radiometric dates from different sites, meaning that ages from well-dated sites can inform age estimates at sites with little or no age information. Age estimates with uncertainty can thus be obtained for any point within any site, and alternative alignments can be identified. Additional stratigraphic knowledge, such as hiatuses or tie points, can be readily incorporated. Our Bayesian model works by evaluating the fit of a single cubic spline (Heaton et al., 2020) to the combined quantitative signal of all sites. If more than one type of signal is included, e.g.  $\delta^{13}$ C and  $\delta^{18}$ O, a different spline is constructed for each signal type, and their joint likelihood is used to evaluate the alignment. Different alignments are generated by shifting the sites relative to each other, and by scaling segments of the sites using different sedimentation rates. Markov chain Monte Carlo methods are used to

obtain the posterior distributions of the unknown model parameters. Our method is implemented as an R package, 'StratoBayes'.

To demonstrate the potential of this method, we apply it to artificial stratigraphic data and to a real case study using lower Cambrian  $\delta^{13}$ C records from Morocco (Magaritz et al., 1991; Maloof et al., 2005, 2010; Tucker, 1986) and Siberia (Kouchinsky et al., 2007). Integrating radiometric dates (Landing et al., 1998, 2021; Maloof et al., 2010), we provide age estimates for the studied sections of an interval spanning several lower Cambrian carbon isotope excursions, and compare our algorithm-derived correlation with recent stratigraphic models relying on visual expert-based interpretations (Bowyer et al., 2022, 2023). Our solution also provides a fully quantitative age estimate for the appearance of the first Siberian trilobites, which are thought to be the world's oldest trilobites (Landing et al., 2021).

## 2 Bayesian stratigraphic model

StratoBayes generates and evaluates alignments of quantitative stratigraphic signals. A signal consists of, for example, geochemical or geophysical measurements that vary across height or depth (Fig. 1a), obtained from a contiguous sedimentary sequence, which may be interrupted by hiatuses at known horizons. Alignments are generated by shifting the sites containing the signals either (a) against a fixed reference site, or (b) against each other on an absolute age scale. Additionally, the sites are scaled ("stretched" or "squeezed") assuming different sedimentation rates. The fit of different alignments, corresponding to different shifts and sedimentation rates, is evaluated in the Bayesian framework.

Statistical analysis in the Bayesian framework starts by formulating a probabilistic model that includes known data y and unknown model parameters  $\theta$ . Instead of trying to identify a single estimate for  $\theta$ , Bayesian inference involves estimating probability distributions for the model parameters, termed "posterior probability distributions". Posterior distributions are obtained by combining prior knowledge of the parameters with the data via a likelihood function. Bayes' theorem states that the probability of the parameters given the data,  $p(\theta|y)$ , i.e. the posterior probability, is proportional to the probability of the data given the model parameters (i.e. the likelihood),  $p(y|\theta)$ , times the prior probability of the model parameters,  $p(\theta)$ :

$$p(\theta|y) \propto p(y|\theta)p(\theta) \tag{1}$$

In our case, this approach requires specifying prior probability distributions for the unknown model parameters that control the shifting and scaling (Fig. 1b), and optionally for the duration of pre-determined hiatuses. Our model assumes that the measurements in each sedimentary sequence are samples (with noise)

from a common underlying signal, whose value can be modelled by a smooth curve described by a cubic B-spline. Our likelihood function quantifies how well a cubic B-spline fitted to a given alignment explains the observed data (Fig. 1c). Additional likelihood components can integrate absolute age constraints such as radiometric dates or other tie points, e.g. index fossils. Using Bayes' theorem, the priors are combined with the likelihood to obtain the posterior probability for any alignment.

We obtain probability distributions for the parameters of the model by running a Markov chain Monte Carlo (MCMC) simulation. This involves repeatedly generating parameter values over a large number of iterations. To ensure thorough exploration of the parameter space, we employ parallel tempering, i.e. we run multiple chains in parallel, flattening the likelihood of the tempered (hot) chains, which can therefore move between different posterior modes; swaps between chains are proposed at every iteration. For the posterior estimates, we retain samples only from the primary (cold) chain. An initial portion of the samples is discarded (burn-in) to remove dependency on starting values, and only every n<sup>th</sup> iteration is recorded to reduce autocorrelation. Details on the MCMC implementation are provided in Appendix A.

In the following, we will assume that measurements were taken on a height scale (increasing from the bottom to the top), but depth-scale measurements can be used interchangeably by inverting their sign.

Figure 1: Schematic of the alignment algorithm. a) Input data: Quantitative stratigraphic measurements (e.g. geochemical data) from two sites recorded along their section height (here given in meters). b) Priors must be placed on the shift parameter  $\alpha$  and on the relative sedimentation rate  $\nu$ . Here,  $\alpha$  determines the reference height (at Site 1) corresponding to the top of the height range of Site 2, and  $\nu$  corresponds to the sedimentation rate of Site 2 relative to Site 1. The vertical, dashed lines denote the  $\alpha$ 

and  $\nu$  values, 12.3 m and 3.0, respectively, that were used in the creation of the data of site 2. c) An alignment corresponding to a single sample from the posterior. The blue dashed line indicates the position of the top of the data from Site 2 at the reference height scale ( $\alpha$ ; median: 12.5 m). The relative sedimentation rate  $\nu$  has been estimated at a median of 2.8, corresponding to a shortening of the dataset from Site 2 relative to the reference site (indicated by the dashed and solid light brown line). Note that the posterior estimates of  $\alpha$  and  $\nu$  are similar, although not identical to the values used in creating the data (see Sect. 3). The curved grey line shows the cubic B-spline corresponding to the alignment.

#### 2.1 Evaluating alignments with a cubic B-spline

- Identifying good alignment positions requires evaluating and comparing different potential alignments. In the Bayesian framework, the measure used for this evaluation is the likelihood. We derive the likelihood of an alignment from its fit to a single cubic B-spline (Eilers and Marx, 1996), fitted to the measurements from all sites, including the reference site (see Fig. 1c).
- We model each measured value  $y_i$  as normally distributed:

$$y_i \sim Normal(\mu_i, \sigma),$$
 (2)

where  $\mu_i$  is the mean, and the standard deviation  $\sigma$  represents the scatter around the spline.  $\mu_i$  is given by the spline function

$$\mu_i = \sum_{j=1}^{k+2} \beta_j B_j(h_i)$$
 (3)

Here,  $\mu$  can be interpreted as an underlying common signal of which the observations from each site, including the reference site, are noisy realisations. k denotes the number of internal knots of the spline, with more knots implying that the spline can potentially capture higher-frequency variations.  $\beta_j$  is the spline coefficient associated with the j-th basis function, and  $B_j(h_i)$  is the j-th B-spline basis function evaluated at a reference height  $h_i$ . A roughness penalty controlled by a smoothing parameter  $\lambda$  is incorporated in the prior on  $\beta$ , such that higher values of  $\lambda$  serve to favour smoother splines (Appendix A). The number of knots and the roughness penalty each influence spline flexibility in different ways: increasing k provides a finer resolution for fitting local features, whereas increasing  $\lambda$  penalizes abrupt changes and yields smoother fits. The knots for the spline can be distributed across the reference height range that the converted measurement heights occupy for a specific combination of shift parameters ( $\alpha$ ) and scale parameters ( $\nu$ ,

i.e. relative sedimentation rates). Our current model implementation uses evenly spaced knots, but knot placement could also follow, for example, the density of measurements. Alternatively, the knots can be fixed at specific heights on the reference scale, in which case combinations of  $\alpha$  and  $\nu$  that result in converted measurement heights falling outside the knot range cannot be evaluated.

The likelihood of an alignment, given  $\beta$ ,  $\sigma$  and  $\lambda$ , is determined by the residual deviations of the  $y_i$  values from the corresponding  $\mu_i$  values. The overall likelihood for n data points is obtained by taking the product over all individual likelihoods for each pair of  $y_i$  and  $\mu_i$ :

$$L(\mathbf{y}|\boldsymbol{\beta},\sigma,\lambda) = \prod_{i=1}^{n} \frac{1}{\sqrt{2\pi\sigma^2}} \times e^{\left(\frac{-(y_i - \mu_i)^2}{2\sigma^2}\right)}$$
(4)

We thus assume that the deviations of the data from the spline are independently and identically distributed 166 according to a normal distribution with mean 0 and standard deviation  $\sigma$ .

Our model allows for using more than one type of measurement simultaneously. In this case, a separate spline is fitted to all data, from all sites, for each type of measurement. The product of all likelihoods from all measurement types gives the overall likelihood.

## 2.2 Alignment and partitioning

In order to generate alignments of stratigraphic signals from different sites, one site is picked as a fixed reference site. The other sites are shifted and stretched (or squeezed) relative to the fixed reference site r. This requires specifying a shift parameter (height)  $\alpha_s$ , which anchors an arbitrary, specified height of site s to a height in the reference site r. Here, we anchor the top of site s, so we set  $\alpha_s = \alpha_{top,s}$  meaning  $\alpha_{top,s}$  will be the height at site r that aligns with the top of site s. To stretch or squeeze site s, a relative sedimentation rate s0 can be specified, where s1 is defined relative to the reference site. For any height s2 at site s3, the corresponding height in the reference site s2 can then be calculated as

$$h_r = \alpha_{top,s} - \frac{1}{\nu_s} \times \left( h_{top,s} - h_{x,s} \right), \tag{5}$$

where  $h_{top,s}$  is the height of the top of site s. Although we here chose the top of site r as the reference horizon  $\alpha$  for simplicity, any horizon at site r can be used as  $\alpha$ . A  $\nu_s < 1$  implies that site s has a lower sedimentation rate than site s, and consequently, s has to be stretched to match s. A s i.e. a higher sedimentation rate at site s will lead to s being squeezed to match s.

The model described here is simple in that the same  $\nu$  is applied to all measurements of the same site. In this scenario, any site may be used as the reference site. Below, we introduce more complex models with more than one sedimentation rate per site, and with hiatuses. With these models, it is practical to select the site with the most sedimentation rate changes and hiatuses as the reference site. This reduces the number of unknown parameters in the model, making it easier to obtain a representative sample from the posterior.

#### 2.2.1 Multiple sedimentation rates per site

- Instead of having one sedimentation rate per site, sites can be partitioned, reflecting for example lithological units, with each partition being modelled with a distinct sedimentation rate:
- $h_r = \alpha_{top,s} \sum_{l}^{n_{p,s}-1} \left( \frac{1}{\nu_p} \times (h_{p,s} h_{p+1,s}) \right) \frac{1}{\nu_{n_{p,s}}} \times (h_{n_{p,s}} h_{x,s}), \quad p = 1...n_{p,s}$  (6)
- Here,  $n_{p,s}$  is the number of partitions encountered from  $h_{top}$  to  $h_{x,s}$ ,  $h_{p,s}$  is the top height of partition p at site s, and  $h_{p+1,s}$  is the top height of the partition below partition p at site s. If  $h_{x,s}$  falls in the first partition
- from the top, the calculation simplifies to the equivalent of Equation 5, with  $h_{P_s}$ , the top height of the first
- partition being also the top height of site s. The relative sedimentation rates of partitions,  $\nu_p$ , can differ for
- each partition in each site, or partitions in different positions within a site or across sites may share
- sedimentation rates.

205

#### 2.2.2 Site-specific sedimentation rate multipliers

The sedimentation rate model above can be further expanded by adding an overall site-specific sedimentation rate multiplier  $\zeta_s$ :

$$h_r = \alpha_{top,s} - \sum_{l}^{n_{p,s}-1} \left( \frac{1}{\zeta_s \nu_p} \times \left( h_{p,s} - h_{p+1,s} \right) \right) - \frac{1}{\zeta_s \nu_{n_{p,s}}} \times \left( h_{n_{p,s}} - h_{x,s} \right), \quad p = 1...n_{p,s}$$
 (7)

- This may be useful in scenarios where sedimentation rates systematically differ between sites, perhaps due
- to varying distances from a sediment source, but where the sedimentation rate ratios of different partitions
- are assumed to be constant across sites.

#### 2.2.3 Hiatuses

- Known hiatuses (also referred to as unconformities or stratigraphic gaps) can be included at specific pre-
- defined locations in a site. Expanding Equation 5 to include gaps of height  $\delta$ , we obtain

$$h_r = \alpha_{top,s} - \frac{1}{\nu_s} \times (h_{top,s} - h_{x,s}) - \sum_{g}^{n_{G_s}} \delta_g , \quad g = 1...n_{G_s}$$
 (8)

- where  $n_{G_s}$  is the number of gaps encountered from  $h_{top,s}$  until height  $h_{x,s}$ . In a correlation on an absolute age scale (Sect. 2.2.5), hiatuses would instead be expressed as durations, not heights.
- **2.2.4** Tie points
- Tie points define specific heights within an aligned site and assign a probability distribution to indicate to
- which horizon these heights correspond on the reference scale. For example, a tie point might be a
- lithological boundary, a biostratigraphic horizon, or a radiometric date. If tie points are specified, the
- likelihood of an alignment is expanded to include not only the fit of the signal data to the spline, but also
- the positions of the ties on the reference height scale relative to the specified probability distribution.
- For example, a point in an aligned section which is tied by observation to the reference section at a position
- $m_t$  with a normally distributed uncertainty with standard deviation  $s_t$  that ends up being shifted to a
- reference height  $h_t$  (computed from the relevant  $\alpha$  and  $\nu$  parameters) contributes a likelihood of

$$L(m_t|h_t,s_t) = \frac{1}{\sqrt{2\pi s_t^2}} \times e^{\left(-\frac{(m_t - h_t)^2}{2s_t^2}\right)}$$
(9)

- to the overall likelihood of the model.
- **2.2.5** Age-scale alignment
- Data on an (absolute) age scale can be aligned using the methods introduced above by using ages instead
- of heights. However, height-scale data can be aligned on an age-scale if absolute age constraints (specified
- as ties) are provided from at least one site. In this case, all sites will be shifted to align on a common age
- scale, i.e., there is no reference site.
- Analogous to the heights in the reference height scale in Equation 5, ages (a) can be calculated as:

$$a = \alpha_{top,s} + \frac{1}{\nu_s} \times \left( h_{top,s} - h_{x,s} \right) \tag{10}$$

- Here,  $\alpha_{top,s}$  is the top age (minimum age), rather than top height (maximum height), of site s. Sedimentation
- rates  $v_s$  need to be expressed on the common age scale, rather than relative to a reference site. Equations
- 6–8 can be modified accordingly for an analysis on the age scale.

It should be noted that due to sedimentation rates being fixed for an entire site or within partitions, our current model implementation does not necessarily result in increasing age uncertainty away from absolute age constraints. Potential sedimentation rate changes within sites or partitions could lead to our model underestimating age uncertainty with growing stratigraphic distance from absolute age constraints (see De Vleeschouwer and Parnell, 2014).

#### 2.3 Priors

The Bayesian framework requires priors to be placed on all unknown model parameters. In our model, these include the alignment parameters (e.g.  $\alpha$ ,  $\nu$ ), the smoothing parameter  $\lambda$ , the residual standard deviation  $\sigma$  (if it is not fixed), and the spline coefficients  $\beta$ . The priors on the alignment parameters determine the range of possible alignments and need to be chosen with care. For the other parameters, weakly informative priors with minimal influence on the analysis are preferred (Appendix A). In addition to those priors, we penalise a lack of overlap by specifying a prior probability of data points from different sites overlapping each other.

#### 2.3.1 Alignment parameters

- The priors on the alignment parameters should reflect the stratigraphic knowledge on the input data. The user may specify different types of prior distributions (e.g., normal, uniform, exponential) for the alignment parameters during model setup.
  - $\alpha$  determines the reference site (site r) height or age that a specific position within the aligned site (site s) corresponds to. In the absence of prior knowledge on how the sites are likely to align, a uniform prior can be placed on  $\alpha$ . For example, if  $\alpha$  refers to the top of site s, a uniform prior on  $\alpha$  with min and max equal to the height or age range of site r implies that the top of site s will be placed within the height range of site r.
  - v is either a relative (height scale alignment) or an absolute (age scale alignment) sedimentation rate. In our model implementation, priors are placed on the natural logarithm of v,  $\ln(v)$ , rather than on v directly. Specifying rate parameters on the logarithmic scale ensures that their priors are symmetric: a doubling or halving of a rate has equivalent distances on the logarithmic scale. If the sedimentation rate is relative,  $\ln(v) < 0$  (i.e. v < 1) results in "stretching", and  $\ln(v) > 0$  (i.e. v > 1) results in "squeezing" of site s relative to site s. In the absence of strong prior knowledge about the relative sedimentation rate, a normal prior on  $\ln(v)$  with a mean of 0 places equal prior probability on "stretching" or "squeezing" of site s relative to site s. The standard deviation requires at least a broad guess of the potential magnitude of sedimentation rate

- differences. For example, a standard deviation of  $\frac{\ln(4)}{1.96}$  places 95% of prior probability on  $\frac{1}{4} < \nu <$ 4 for  $\ln(\nu) \sim Normal\left(0, \frac{\ln(4)}{1.96}\right)$ . If  $\nu$  is an absolute sedimentation rate, the range of plausible prior sedimentation rates may be estimated from the absolute age constraints.
- $\zeta_s$  is a multiplier applied to all relative or absolute sedimentation rates  $\nu$  corresponding to a single site s. As with  $\nu$ ,  $\ln(\zeta_s) < 0$  (i.e.  $\zeta_s < 1$ ) causes additional "stretching", and  $\ln(\zeta_s) > 1$  (i.e.  $\zeta_s > 0$ ) causes additional "squeezing" of site s.
  - δ is the reference height range or duration of a hiatus. An exponential prior may be useful when little is known about the extent of the hiatus, placing higher probabilities on short extents. The rate needs to be chosen to make sense in the context of the height of the sections, or of the anticipated age range of the sites.

#### 2.3.2 Penalising a lack of overlap

- Individual splines fitted to data from each site separately can almost always follow the data more closely than a single spline fitted to aligned data from all sites. Given enough knots, alignments in which the data do not overlap, or only overlap little, will thus generally result in a higher likelihood than alignments with a partial or full overlap. This means that if the priors allow non-overlapping alignments, those will generally be preferred in the model inference. To counteract this tendency, we impose a prior on the overlap of each individual data point from all sites that penalises non-overlap with data from other sites.
- The prior on overlap for data point i from site s is

$$P(i_s) = e^{\left(-\sqrt{S-1} + \sqrt{S_{overlap,s,i}}\right) \times c_{overlap}}, \quad (11)$$

where S is the number of sites in the analysis,  $S_{overlap,s,i}$  is the number of other sites overlapping the reference height  $h_r$  or age a of point  $i_s$ , and  $c_{overlap}$  is a constant. This formulation implies that the penalty for a point  $i_s$  that overlaps all other sites is 0, and the penalty is strongest (most negative) if  $i_s$  overlaps no other sites. To work effectively, the penalty needs to be stronger for data sets with little noise (low residual  $\sigma$ ), to offset the larger likelihood differences resulting from fitting a spline with low  $\sigma$ . A range of  $c_{overlap}$  values may work in practice. A formulation that we have found works well in many scenarios sets

$$c_{overlap} = c \times \frac{1}{S} \sum_{s=1}^{S} \left( \frac{\sigma_{y,s}}{\sigma_s} \right)^q$$
 (12)

where c is a constant determining the strength of the overlap penalty (set to a default of  $c = \frac{1}{4}$ ), q = 1 if  $\sigma$  is fixed, and  $q = \frac{1}{2}$  if  $\sigma$  is variable (i.e. estimated in the model inference). Here,  $\sigma_{y,s}$  is the standard deviation of all data y from site s, and  $\sigma_s$  is the residual standard deviation of a Bayesian spline fitted to the data y from site s, using the same priors as for the overall model inference.

## 3 Model illustration

- We illustrate the performance of our stratigraphic alignment method with a simple, artificial dataset (Fig.
- 2a). We generated measurements from a reference site (Site<sub>ref</sub>) using a sine wave covering 3.5 periods,
- where each period corresponds to  $2\pi$  radians. To generate the signal data, we intercepted this sine wave at
- heights h with 250 evenly spaced points per period, i.e. the number of data points (n) is  $3.5 \times 250 = 875$ .
- Each signal value  $y_i$  was generated with random white noise  $\sigma = \frac{1}{5}$  added, such that

$$y_i \sim Normal\left(\eta_i \sin\left(h_i - \frac{1}{2}\pi\right), \sigma\right), \quad i = 1...n$$
 (13)

- The factor  $\eta_i$  modulates the amplitude of the sine wave at each height  $h_i$ . It was set to  $\eta = 1$  for the heights
- ranging from  $-0.5\pi$  to  $5\pi$ , and to  $\eta = 0.75$  from heights  $5\pi$  to  $6.5\pi$ , which reduces the amplitude
- beginning in the middle of the third period of the sine wave. The aligned signal was simulated as above,
- but from a sine wave covering one period, sampling 250 data points, again with random noise using  $\sigma = \frac{1}{5}$
- and  $\eta = 1$ . To simulate a sedimentation rate twice as high as at the reference site, we multiplied the heights
- of Site<sub>align</sub> by 2. The heights of Site<sub>align</sub> were then shifted to start at 0.
- The aligned signal should thus match either the first or the second, but not the third period of the reference
- signal. To align the two sites, we used a simple model with a site-specific shift  $\alpha$ , referring to the top of
- Site<sub>align</sub> and relative sedimentation rate  $\nu$  as in Equation 5. From the data generation, we know that the
- posterior of  $\nu$  should be  $\approx$  2, with  $\ln(\nu) \approx$  0.69, and  $\alpha$  (defined as the reference height corresponding to
- the top height of Site<sub>align</sub>) should be  $\approx 2\pi$  (top of first period) or  $\approx 4\pi$  (top of second period).
- To minimise the influence of the priors, we used a uniform prior on  $\alpha$  that extends well beyond the
- alignment positions known from generating the data, and a broad normal prior on ln(v) that encompasses
- the known sedimentation rate v = 2 (Fig. 2b):

$$P(\alpha) \sim Uniform(-\pi, 8\pi) \qquad (14)$$

 $P(\ln(v)) \sim Normal(0,1)$ 314 (15)315 These priors place 95% of prior probability for the relative sedimentation rate of Site<sub>align</sub> between 0.14 and 316 7.1, and place the top of Site<sub>align</sub> anywhere from half a period below the start of the first period  $(-\pi)$  up to one period above the third period  $(8\pi)$ . For the cubic spline, we specify 20 evenly spaced knots, which is 317 318 more than enough to approximate the three periods of the sine wave. 319 We estimated the posterior of the model with three independent runs, each with 16 chains and 60,000 iterations. The first 10,000 samples were discarded as burn-in, and every 25th iteration was recorded, 320 321 resulting a total of 6000 samples after burn-in across all three independent model runs. 322 The results show that the analysis identified both matching alignments, corresponding to the first and 323 second period of the reference site (Fig. 2b). The posterior probability for (Site<sub>align</sub>) matching period 1 is 50.1%, and 49.9% for matching period 2. A density plot of the posterior of  $\alpha$  and  $\ln(\nu)$  shows that  $\alpha$  has a 324 325 bimodal posterior, corresponding to the two alignments (Fig. 2c). The trace plots indicate good mixing of 326 the chains (Fig. 2d), suggesting that the posterior estimates are robust. 327 It is notable that the model estimate for the relative sedimentation rate  $\nu$  is lower at 1.90 (95% credible 328 interval: 1.82 to 1.99) than the value used for the data generation (2.00). Reported values, here and 329 throughout, represent the posterior median, with 95% credible intervals – given in brackets – referring to 330 the interval between the 2.5% and 97.5% points of the posterior distribution. This deviation of the posterior 331 from the known sedimentation rate estimate arises because the priors favour greater overlap (see Sect. 332 2.3.2). The posterior alignment tends to "compress" the data from Site<sub>align</sub> slightly less than expected, 333 leading to an increased overlap of points (see also Fig. 5b).

Figure 2: Model illustration using artificial data. a) Input data: Quantitative stratigraphic data from two sites. The blue line indicates the range in which  $Site_{ref}$  was created with  $\eta=1$ , and the purple line above indicates the range for which  $\eta$  was set to 0.75 to lower the amplitude. b) Two alignments identified by the inference, with (Site<sub>align</sub>; blue squares) matching the first or second period of (Site<sub>ref</sub>; red points). The alignments shown here correspond to two distinct samples from the posterior; other samples will result in slightly different positions of (Site<sub>align</sub>). The curved dark lines show the cubic spline corresponding to each alignment. c) Posterior densities of  $\alpha$  and  $\ln(\nu)$ . The two modes of  $\alpha$  correspond to the two distinct alignments in b). The dotted lines indicate the  $\nu$  values with which

(Site<sub>align</sub>) was simulated, and the two plausible  $\alpha$  values. d) Trace plots of  $\alpha$  and  $\ln(\nu)$ . The three distinct colours correspond to the three independent model runs. For visual clarity, only 75 selected samples are shown from each run.

## 4 Case study: Lower Cambrian δ<sup>13</sup>C records

- To demonstrate the utility of this method, we use it to align stable carbon isotope records ( $\delta^{13}$ C) from lower Cambrian marine shelf carbonates (Fig. 3). We integrate a combination of radiometric dates,  $\delta^{13}$ C and astrochronological information from four sites to obtain age estimates for the sampled intervals from all sites, and use this age model for dating the first documented occurrence of Siberian trilobites.
  - 4.1 Data

We selected three records from the Anti-Atlas mountains in Southern Morocco, corresponding to the Oued Sdas, the Tiout and the Talat n' Yissi sections, which were part of West-Gondwana during the early Cambrian (Magaritz et al., 1991; Maloof et al., 2005, 2010; Tucker, 1986). Oued Sdas and Tiout harbour multiple precise U-Pb radiometric ages (Landing et al., 2021; Maloof et al., 2010). Talat n' Yissi has no radiometric dates, but a radiometric date exists from the stratigraphically equivalent Lemdad syncline (Landing et al., 1998) that has been correlated biostratigraphically to Talat n'Yissi with the *Antatlasia gutta-pluviae* zone (Maloof et al., 2005); we include this date in the analysis. We will align these sites with each other, and with a  $\delta^{13}$ C record from the Sukharikha section from the northwestern Siberian platform (Kouchinsky et al., 2007), corresponding to the palaeocontinent Siberia. There are no radiometric dates available for the Siberian section for this stratigraphic interval. Data that was inferred to be below the lower leg of the prominent "5p" excursion (lowest peak in Fig. 3a and d) was excluded to simplify the correlation, reducing the number of modelled sedimentation rates unconstrained by radiometric dates. This cropping of data affects the Oued Sdas and Sukharikha sections; Fig. 3 shows all data that was included in the analysis.  $\delta^{13}$ C values were used as reported by the authors of the respective publications without any scaling or other adjustments.

Figure 3: Cambrian  $\delta^{13}$ C data and radiometric dates from Morocco (a - c) and Siberia (d). Different colours, in conjunction with different symbols, delineate different lithological units or formations. Circles indicate the position of radiometric dates, with mean age and 2 standard deviations denoting the uncertainty. Stars denote the positions where the oldest trilobite remains are found in Morocco (a) and Siberia (d). The dashed line in (d) indicates a hiatus.  $\alpha$  indicates the reference horizon chosen for specifying the prior on the shift parameter  $\alpha$  for each site.

## 4.2 Model specification

To align the four sites on the age scale, we specify an  $\alpha$  parameter on the absolute age scale (Ma) for each site, and use absolute, rather than relative sedimentation rates (expressed in m Myr<sup>-1</sup>). We encapsulate variation in sedimentation rates ( $\nu$ ) by partitioning sites into members, formations or lithological units, leading to multiple sedimentation rates per site. As there are few radiometric dates to constrain sedimentation rates, partitions shared between the Moroccan sites are set to have the same relative sedimentation rate across sites. To account for potentially faster or slower sedimentation rates at different sites, a site-specific sedimentation rate multiplier  $\zeta$  is added for Oued Sdas and Talat n'Yissi that is multiplied with the  $\nu$  from those sites. The  $\nu$  for a partition applies to all sites at which this partition occurs; for Tiout, they are used unaltered, and no  $\zeta$  is needed for Sukharikha as there are no shared partitions with other sites. We partition the Moroccan data based on the lithostratigraphy from Maloof et al. (2005). We divide the Adoudounian Tifnout Member into a lower part (Tifnout 1.), and an upper stromatolitic part (Tifnout stromatolite), as preliminary results suggested pronounced sedimentation variability between those parts. We subdivide the Lie de Vin Formation into three members; the Igoudine Formation is subdivided into two members. The Amouslek and Isaafen formations are not subdivided. The Sukharikha section is

divided into two formations, which we assign separate sedimentation rates. At the boundary, a substantial hiatus is evinced by the truncation of the "7p"  $\delta^{13}$ C peak (Kouchinsky et al., 2007). We include the duration of this hiatus ( $\delta$ ) as an additional unknown parameter in the model.

The model requires priors to be specified for each of its 18 alignment parameters: Four  $\alpha$ , eleven  $\nu$ , two  $\zeta$  and one  $\delta$  (Fig. 4). These priors are broadly guided by the radiometric dates and by previous work (Bowyer et al., 2023; Landing et al., 2021; Sinnesael et al., 2024). The  $\alpha$  for the Tiout and Sukharikha sites are placed at the height positions of the first trilobite fossil remains found at Tiout (Sinnesael et al., 2024), and the first appearance of Siberian trilobites correlated to Sukharikha (Landing et al., 2021; Varlamov et al., 2008). Here, we place normal distributed priors with mean age 520 Ma and a wide standard deviation of 2 Myr on the  $\alpha$  parameters at Tiout and Sukharikha. This prior reflects the notion that first appearance dates of trilobites may be broadly similar at  $\approx 520$  Ma, but not necessarily identical, and the data is allowed to determine the exact age of each  $\alpha$ . The  $\alpha$  priors for Oued Sdas and Talat n'Yissi are placed at the position of the lowest or the only available radiometric date, respectively, consisting of normal distributions with mean age equal to the mean age estimate of the radiometric data and a wide standard deviation of 2 Myr.

For the sedimentation rates, priors informed by an astrochronology of the Tiout section (Sinnesael et al., 2024) are used for the following five stratigraphic partitions: The lower, middle and upper members of the Lie de Vin Formation, and for the lower and upper (Tiout Member) members of the Igoudine Formation. Those priors are chosen such that the 95 percentile interval of  $\nu$  spans the minimum and maximum of the astrochronological sedimentation rate estimates when using an uncertainty of  $\pm 1$  short eccentricity cycle for each partition, with an estimated duration of short ( $\approx 100 \, \text{kyr}$ ) eccentricity cycles ranging from 92.5 to  $100.5 \, \text{kyr}$  (two standard deviations, following Lantink et al., 2022).

To specify priors for the remaining Moroccan partitions (lower part of Tifnout Fm., Tifnout stromatolite, Amouslek Fm., and Isaafen Fm.), sedimentation rates between the radiometric dates from Oued Sdas and Tiout are calculated using the mean ages of the dates. The prior on  $\ln(\nu)$  is defined as a normal distribution with a mean of 5.39, corresponding to the mean of the empirical sedimentation rates from Oued Sdas and Tiout, calculated on the logarithmic scale. A wide standard deviation of 0.75 is set, resulting in the 95 percentile interval of  $\nu$  spanning 50.3 to 951 m Myr<sup>-1</sup>. This interval significantly exceeds the range of sedimentation rates inferred from the radiometric dates at Oued Sdas and Tiout, 147 to 314 m Myr<sup>-1</sup>, allowing for the possibility of lower or higher sedimentation rates in some partitions.

Prior sedimentation rate estimates for the Siberian formations are estimated in the absence of radiometric dates, very broadly based on global correlations by Bowyer et al. (2023). These correlations suggest average sedimentation rates on the order of 20 to  $30 \,\mathrm{m\,Myr^{-1}}$ ; we place a normal prior on  $\ln(\nu)$  with a mean of

402 3.30 and a standard deviation of 0.75, resulting in a 95 percentile interval of  $\nu$  spanning 6.23 to 403 117.9 m Myr<sup>-1</sup>, which allows for the possibility of significantly different sedimentation rates from those inferred by Bowyer et al. (2023). 404 405 Finally, a prior needs to be placed on the duration of the hiatus  $\delta$  between the Sukharikha and the 406 Krasnoporog formations. Kouchinsky et al. (2007) do not give an indication of the potential duration of this hiatus, but if the under- and overlying  $\delta^{13}$ C peaks are correlated as indicated by previous work (Bowyer et 407 408 al., 2022; Landing et al., 2021), a relatively short hiatus of  $\approx 1$  Myr is likely. To express considerable 409 uncertainty about the duration of the hiatus, we place an exponential prior on  $\delta$  with a rate of 1, which 410 places 95% of prior probability on the duration being 

Figure 4: Priors on the 18 alignment parameters for the Cambrian model. Prior probability density is shown (a) for four  $\alpha$  parameters corresponding to one site each (priors for Tiout and Sukharikha in grey are identical), (b) for six  $\nu$  (sedimentation rate) parameters with little prior knowledge, (c) for five  $\nu$  parameters from Morocco with tight priors based on astrochronology, (d) for  $\zeta$  parameters (site-specific sedimentation rate multipliers) for Oued Sdas and Talat n'Yissi (identical), and (e) for the duration of the hiatus between the Sukharikha and the Krasnoporog formations. The width of the red bar in (b) visualises the range of sedimentation rates spanned by (c). Panel (f) visualises two alignments generated by randomly drawing parameter values from their respective priors, to give an indication of the broad range of alignments that the priors on the alignment parameters allow; colours correspond to the four sites (see Fig. 6). Panels (b), (c), and (d) are depicted with a logarithmic x-axis as the priors were specified on  $\ln(\nu)$  and  $\ln(\zeta)$ .

#### 4.3 Parameter estimation

This model is more complex than our earlier examples, and hence requires longer runs with more chains. We conducted four independent model runs, each with 750,000 iterations and 24 chains. The runs were executed in parallel using four workers on a desktop computer (Intel i7-10700 CPU, 8 cores / 16 threads, 40 GB RAM) and completed within 5 days. The first 150,000 iterations were discarded as burn-in. From

the remaining 600,000, every  $50^{th}$  iteration was retained, resulting in 12,000 samples per run and 48,000 samples in total.

Inspection of trace plots of the model runs indicates stationarity and good mixing of the chains with the exception of infrequent visits of secondary posterior modes (Appendix B, Fig. B1). The potential scale reduction factor (using eq.4 in Vats and Knudson, 2021) is between 1.00 and 1.05 for all alignment parameters, suggesting approximate convergence of the MCMC. The multivariate effective sample size (Vats et al., 2019) of the 48,000 samples is 4161.

#### 4.4 Results

To identify distinctly different alignments in the posterior, a hierarchical density-based cluster analysis (Campello et al., 2015) was conducted using the inferred ages of all partition boundaries of the four sites (Fig. 4a,b). We specified 1% of samples (480) as the minimum number of points per cluster, resulting in three distinct clusters with 93%, 2.8% and 2.6% of posterior samples, respectively, and 1.5% of samples not being assigned to any cluster. These alignment clusters also differ in the prior probabilities and likelihoods associated with individual posterior samples. On average, samples from alignment 1 tend to exhibit a lower degree of overlap, but a higher likelihood (Fig. 4c), indicating a better fit to the data.

Figure 5: (a, b) 2D density plots of the inferred top ages of the four sites, representing some of the ages used for obtaining alignment clusters from posterior samples. (c) 2D density plot of the ln prior probability of overlap against the overall ln likelihood. Areas with more opaque shadings correspond to a higher density of individual posterior samples. Colours correspond to alignment clusters: alignment 1 - violet; alignment 2 - blue; alignment 3 - green; outlier samples not assigned to any cluster - yellow.

443444

447448

458459

2, and 0.63 Myr (0.53 to 0.74 Myr) for alignment 3.

Using samples from the posterior of the model parameters, alignments can be generated. Fig. 6 visualises three alignments drawn from the three alignment clusters identified in the posterior. For each alignment cluster, the iteration with partition boundary ages that are, on average, closest to the median ages of the partition boundaries within that cluster is selected for displaying. All three alignments exhibit a good match between the long-term trends of the  $\delta^{13}$ C curves from the four sites and the common spline curve, although many shorter-term deviations are visible (Fig. 6a-c). The spline curve notably follows the more densely sampled sites (Oued Sdas, Talat n'Yissi) more so than the thinly sampled sites (Tiout, Sukharikha), resulting in greater deviations of the latter two sites. The posterior age estimates for the stratigraphic positions of the radiometric dates broadly match the age estimates that were used as inputs in the analysis (Fig. 6d). The deviations are greatest for the Talat n'Yissi date (Ta<sub>1</sub>), which has large uncertainty and therefore less influence on the analysis, and the second date from Oued Sdas (Ou<sub>2</sub>). The first appearances of trilobites are visualised alongside the dates in Fig. 6d, and are dated to 519.46 Ma (519.25 to 519.68 Ma) at Tiout. The age estimate for the first Siberian trilobites differs considerably between the different alignment solutions: For the most likely alignment 1, the age estimate is 520.79 Ma (520.98 to 520.61 Ma), and for alignment 2 the estimate is somewhat higher at 521.05 Ma (521.19 to 520.91 Ma). Alignment 3 suggests a significantly later appearance of Siberian trilobites at 519.98 Ma (520.15 to 519.84 Ma). All three alignments place the appearance of the first Siberian trilobites before their appearance at Tiout, with the temporal gap (computed directly from the posterior distribution) being estimated at 1.33 Myr (1.09 to 1.54 Myr) for alignment 1, 1.71 Myr (1.54 to 1.87 Myr) for alignment

Figure 6: Three possible alignments identified by the inference with Cambrian data. (a) Exemplary sample from the cluster of the most likely alignment (93% of posterior samples). (b, c) Exemplary samples from a second and third identified alignment cluster (2.8% and 2.6% of posterior samples, respectively). Each shown alignment corresponds to a single sample from the posterior; other samples will result in slightly different alignments. 1.5% of samples were not assigned to any cluster (see Fig. 5). The curved dark lines show the cubic B-splines corresponding to each visualised sample. The coloured bars to the right of each alignment show the median duration of the stratigraphic partitions under each respective alignment cluster, based on the median ages of partition boundaries, with colours repeating the colour scheme of Fig. 2. (d) Posterior density of the inferred ages corresponding to the radiometric dates to the left (3 from Tiout, 4 from Oued Sdas, and 1 from Talat n'Yissi) and the first occurrences of trilobites at Tiout (Ti tril.) and Sukharikha (Sh tril.) to the right, in colours. All samples from all alignment clusters were included. Greater width corresponds to higher posterior density; all densities are scaled to have the same maximum for better visibility. Densities representing the uncertainties of radiometric dates based on their mean and standard deviation are shown in grey (left). The faint yellow shading to the right shows the prior density on  $\alpha$ , i.e. the first appearance of trilobites at Tiout and Siberia based on a mean age of (520 Ma) and a standard deviation of 2 Myr (identical for

Tiout and Siberia). Colours and shapes of the points correspond to the four sites: Tiout - brown circles; Oued Sdas - pink squares; Talat n'Yissi - green diamonds; Sukharikha - blue triangles.

The posterior of the model runs allows the construction of age models that span the entire height of each site (Fig. 7). As sedimentation rates are constrained to be constant within the pre-defined partitions, sedimentation rate changes are visible as inflections at the boundaries of these partitions. Age uncertainties are relatively low at Tiout and most of Oued Sdas, which are relatively well constrained by radiometric dates in the top (Tiout) and middle (Oued Sdas) parts of the sections, as well as by astronomical priors on sedimentation rates. Uncertainty noticably increases towards the top and bottom of Oued Sdas. The lowest partition of Oued Sdas is constrained only by its match to the lower part of the Sukharikha Fm., their age estimates are thus varying considerably between different alignments (Fig. 6). Differences in the positioning of the  $\delta^{13}$ C curves between alignments are greatest at Talat n'Yissi and the Siberian Krasnoporog Fm. (Fig. 6), which results in large uncertainties in the age models (Fig. 7c, d).

Figure 7: Age-depth model for each of the four sites. The solid lines indicate the median posterior ages corresponding to the respective heights; the shaded interval denotes the 95% credible interval of posterior ages. Colours correspond to the colours of partitions introduced in Fig. 3. Circles indicate the mean age estimates of radiometric dates, with vertical lines spanning two standard deviations around the mean of these age estimates. Stars denote the first appearances of trilobites in Morocco and Siberia. See Fig. B2 for separate visualisations of age-depth models for different alignment solutions.

We used StratoBayes to correlate and date four lower Cambrian carbonate sections using  $\delta^{13}$ C records,

## **5 Discussion**

474475

486

## 5.1 Lower Cambrian stratigraphy

radiometric dates and astrochronological sedimentation rate estimates. From a large space of possible alignment configurations (Fig. 4), the software identified alignment solutions that visibly match the largescale features in the  $\delta^{13}$ C records from multiple sites, while simultaneously achieving an approximate fit to the radiometric dates (Fig. 6). The most likely alignment solution from the posterior, alignment 1 (probability = 93%), results in a correlation of the three Moroccan sites that has much in common with that proposed by Maloof et al. (2005). In our model, we used common sedimentation rates for the stratigraphic partitions (members, formations) shared between the sites, whilst allowing sedimentation rates to systematically differ from the reference sedimentation rates at Tiout by adding a site-specific multiplier. This multiplier,  $\zeta$ , is 1.02 (95% credible interval: 0.97 to 1.08) for Oued Sdas, meaning the model estimates very similar sedimentation rates for Tiout and Oued Sdas (Fig. 6a), consistent with their close geographical proximity. Sedimentation rates for the shared partitions at Talat n'Yissi are lower by a factor of 0.86 (0.76 to 0.96), which would be consistent with a moderately lower accommodation space at Talat n'Yissi relative to Tiout and Oued Sdas (as suggested by Fig. 3B in Maloof et al., 2005). We deliberately chose broad priors that did not explicitly enforce a relationship between sedimentation rates and palaeogeography; nonetheless, the model identified a geologically plausible solution. In contrast, the higher  $\zeta_{Talat\ nvYssi}$  of alignment 2 (probability = 2.8%, 1.07 to 1.37) and alignment 3 (probability = 2.6%, 2.07 to 2.45) are harder to reconcile with the palaeogeographic context. Alignments 2 and 3 also suggest different sedimentation rates between Tiout and Oued Sdas, with a higher

value of  $\zeta_{Oued\ Sdas}$  (1.13 to 1.26) being estimated by alignment 2, and a lower value of  $\zeta_{Oued\ Sdas}$  (0.83 to

495 0.88) by alignment 3. The most consistent lithostratigraphic alignment between Tiout and Oued Sdas is 496 achieved by alignment 1, meaning that the age estimates for partition boundaries (based on members or 497 formations) are most similar (Fig. 6). For the more distant Talat n'Yissi, age estimates of partition 498 boundaries differ to varying degrees across all three alignments.

Breaking down the posterior probability into individual components – likelihood (fit of  $\delta^{13}$ C measurements to the spline, fit of age estimates to the radiometric dates) and prior probability from the overlap penalty – reveals that samples from alignment 1 have a higher likelihood, on average (Fig. 5c). In contrast, alignments 2 and 3 have a greater number of overlapping  $\delta^{13}$ C points, which results in higher overlap prior probabilities (Fig. 5c). The overlap prior reflects the prior belief that substantial parts of the sections involved in the correlation should be overlapping. However, the weight of that prior is somewhat arbitrary and reflects the technical requirement to facilitate overlap despite non-overlap allowing for closer fit to the spline, similar to the role of the "edge value" in some DTW implementations (Hay et al., 2019). A lower prior weight on overlap would thus have caused alignments 2 and 3 to receive lower posterior probabilities relative to alignment 1. Taken together, the evidence from above leads us to strongly favour alignment 1, and we will focus further discussion on that most likely alignment solution.

A radiometric date of  $517.0 \, \text{Ma}$  ( $\pm 2 \, \text{SD}$ :  $515.5 - 518.5 \, \text{Ma}$ ) has been recovered from the Lemdad Syncline in the Atlas mountains (Landing et al., 1998), and has been correlated biostratigraphically to a horizon in the lower Isaafen Fm. at Talat n'Yissi (Maloof et al., 2005). In our alignment 1, this horizon has a posterior age estimate of  $519 \, \text{Ma}$  ( $519.2 \, \text{to} 518.8 \, \text{Ma}$ ) –  $\approx 2 \, \text{Myr}$  older than the mean of the radiometric date. This date has informed the age estimates for Talat n'Yssi in Maloof et al. (2005) and Maloof et al. (2010), whereas alignment 1 produces age estimates close to those of Bowyer et al. (2022) and Bowyer et al. (2023). Age estimates deviating from radiometric dates are not necessarily incorrect: Although radiometric dates are sometimes treated as "absolute truth" within the stratigraphic community, they are the result of various sources of technical uncertainties (Condon et al., 2024) and geological interpretations like the actual zircon crystallisation versus eruption age (Keller et al., 2018). This is illustrated by the recalculation of the radiometric date from Landing et al. (1998) to  $515.56 \, \text{Ma}$  ( $\pm 2 \, \text{SD}$ :  $514.40 - 516.72 \, \text{Ma}$ ) in the Geological Time Scale 2012 (Schmitz et al., 2012).

The two radiometric dates measured at Tiout at the bottom of and within the Amouslek Formation suggest a sedimentation rate of  $146 \,\mathrm{m\,Myr^{-1}}$  ( $\pm 2\,\mathrm{SD}$ : 78.7 to  $613 \,\mathrm{m\,Myr^{-1}}$ ) for the Amouslek formation. However, the posterior estimates for the sedimentation rate in the Amouslek formation are poorly constrained and high compared to the sedimentation rates of all other partitions, at  $3030 \,\mathrm{m\,Myr^{-1}}$  ( $800 \,\mathrm{to} \,17,300 \,\mathrm{m\,Myr^{-1}}$ ). It appears that the model has overestimated the Amouslek sedimentation rate in aligning the  $\delta^{13}\mathrm{C}$  record of

the overlying Isaafen formation with a part of the Siberian Krasnoporog formation which has similar  $\delta^{13}$ C values (Fig. 6a). The alignments of Bowyer et al. (2022) imply significant sedimentation rate changes within the Krasnoporog formation, allowing the  $\delta^{13}$ C records to be better reconciled with the radiometric dates. We didn't allow for sedimentation rate changes within the Krasnoporog formation because the stratigraphic log of Kouchinsky et al. (2007) indicates a uniform facies. Additional sedimentation rate changes might lead to a closer alignment with the radiometric dates, at the cost of greater model complexity. The alignment of the Siberian Sukharikha section with the Moroccan sites is relatively precise in the lower half of the records: The prominent positive  $\delta^{13}$ C excursions interpreted as the "5p" and "6p" excursions have a similar magnitude both at Oued Sdas and Sukharikha, and are readily aligned visually (Bowyer et al., 2022) and by our model (Fig. 6). Our model aligns the main 6p peak of Sukharikha with the first subpeak of the second large excursion at Oued Sdas, as in model C in Bowyer et al. (2022). The lesser, positive excursion below the hiatus at the top of the Sukharikha formation lines up with the positive excursion in the lower Lie-de-Vin formation, representing the "II" peak as in model C in Bowyer et al. (2022). The upper parts of the Moroccan records and the Siberian Krasnoporog formation appear to be aligned primarily by matching the prominent positive excursion interpreted as excursion "IV" (Bowyer et al., 2022; Kouchinsky et al., 2007). The "III" peak below is only weakly expressed at Oued Sdas, leading to uncertainty in the alignment with the corresponding part of the Krasnoporog formation, and in the inferred duration of the hiatus even within alignment solution 1 (Fig. B3a-c). Similarly, considerable uncertainty exists in how the top of Talat n'Yissi corresponds to the Krasnoporog formation. This is evident from variations between samples in alignment solution 1 (Fig. B3a-c) and in the wide credible intervals of those parts of the age models (Fig. 7). The relatively small magnitude of  $\delta^{13}$ C changes limits the model's ability to identify a definitive alignment solution for that part of the record. Our estimate for the Moroccan first appearance of trilobites at Tiout from alignment 1, 519.47 Ma (519.68 to 519.26 Ma), is slightly younger and somewhat less precise than the recent, astrochronological estimate of 519.62 Ma (95% highest posterior distribution: 519.70 to 519.54 Ma) by Sinnesael et al. (2024). We attribute this difference to our model simultaneously combining different data types from multiple sites. Additionally, Sinnesael et al. (2024) allowed sedimentation rates to vary between cycles, whereas our model assumed a single sedimentation rate per member. In our alignment 1 solution, the highest  $\delta^{13}$ C values of Tiout correlate to shortly after the peak of the IV  $\delta^{13}$ C excursion. This correlation suggests that the actual peak of the excursion at Tiout has not been sampled by Magaritz et al. (1991) and Tucker (1986), which may result in misalignments when correlating the record to other sections. Further  $\delta^{13}$ C samples from the lower Igoudine and upper Lie-de-Vin formation at Tiout are required to improve correlation with other

534535

556557

sections, including the correlation presented herein.

Our model successfully reconstructs the first appearance of trilobites at Tiout, within error, despite using a simpler astrochronology and enforcing a less variable sedimentation rate history than Sinnesael et al. (2024). It also provides the first fully quantitative estimate for the first appearance of trilobites in Siberia based on chemostratigraphic correlation and the Moroccan radiometric dates and astrochronology, at 520.79 Ma (520.98 to 520.61 Ma). This refines earlier estimates of  $\approx$  521 Ma (Landing et al., 2021), and quantifies the temporal gap between the appearance of trilobites in Siberia and Morocco as 1.33 Myr (1.09 to 1.54 Myr). We do not suggest that these estimates are definitive; indeed, we anticipate that the incorporation of additional  $\delta^{13}$ C data from Tiout, the inclusion of astrochronological estimates of individual short eccentricity cycles, and the relaxation of the assumption of constant sedimentation rates within partitions may update the estimate. A high-resolution temporal sequence of trilobite first occurrence dates could be used to delineate trilobite evolutionary rates and dispersal; to evaluate evolutionary hypotheses on the origins and biomineralisation of trilobites (Holmes and Budd, 2022; Paterson et al., 2019); and to inform the definition of the base of the Cambrian Series 2 (Zhang et al., 2017).

#### 5.2 Statistical alignment and age modelling

#### 5.2.1 Advantages of Bayesian stratigraphic alignment

- As shown above, our algorithm can identify the correct alignment positions in scenarios with one (Fig. 1)
- or more than one (Fig. 2) known solution. In scenarios where more than one distinctly different alignment
- is identified, the probability of each solution, given the specified data and model, is identified. This can be
- used to evaluate the likelihood of competing models for the alignment of stratigraphic records found by
- visual (e.g. Bowyer et al., 2023; Landing and Kruse, 2017) or algorithmic (e.g. Hay et al., 2019) correlation.
- The requirement to specify priors for the alignment parameters can be leveraged to provide information
- beyond that which is contained in the signals: for example, information on sedimentation rates may be
- expressed in the prior.
- Because our model can integrate absolute age constraints such as radiometric dates, a user is able to
- correlate stratigraphic records and construct probabilistic age models in a single step. In our Cambrian
- example, the posterior alignment and the posterior age model are thus influenced by the priors, the
- quantitative signals and the radiometric dates. In contrast, age models constructed in a separate step after
- identifying alignments do not reflect uncertainty arising during the alignment stage (Hagen and Creveling,
- 2024).

568569

- In our integrated approach, discrepancies between radiometric dates and signal alignment are resolved
- probabilistically, with the model weighting the available evidence based on its likelihood and prior

information. This means that posterior age estimates may diverge from the age information provided by radiometric dates, as seen with the Ou<sub>2</sub> date in Fig. 6d. This is not necessarily a deficiency of the model; rather, it indicates that the priors and non-radiometric data provide sufficiently strong evidence to suggest that the actual age of the horizon associated with the radiometric date falls toward the tails of its confidence interval, or that the radiometric uncertainty may be underestimated. Some degree of discrepancy is expected when integrating multiple data types rather than relying on a single proxy (see also Lee et al., 2022).

If, on the other hand, the user wishes to increase the influence of radiometric dates on the posterior age estimates, this can potentially be achieved by introducing additional sedimentation rate changes to allow more flexible alignment of the proxy signals, reducing the weight of the proxy signal records – such as by imposing a larger  $\sigma$  for the cubic spline – or by weakening priors.

#### 5.2.2 Model choice and priors

Stratigraphic alignment using algorithms has the advantage of removing some of the inherent subjectivity of visual alignment (Sylvester, 2023). Yet, somewhat subjective decisions are still explicitly or implicitly made with every alignment algorithm. In the case of DTW, subjectivity is introduced e.g. with restrictions on the warping path (i.e. relative sedimentation rates, Sakoe and Chiba, 1978), with the amount of overlap required between sections (Hay et al., 2019), or with the choice of an exponent controlling the weight of outlier values (Wheeler and Hale, 2014). All of those settings can alter the outcome of DTW-based alignments. Likewise, our Bayesian approach comes with a number of subjective choices. The appropriate model structure can be readily determined when the data-generating process is known (Sect. 3), but has to be carefully considered and potentially revised when dealing with complex real-world data (Sect. 4). Lithological data may guide the partitioning of data and can inform somewhat objective choices of horizons with likely sedimentation rate changes (Sect. 4.2), but such information may not be readily available with some datasets, such as with well logs.

Besides the model structure, StratoBayes requires the user to specify priors for several model parameters: relative or absolute sedimentation rates  $(\nu, \zeta)$ , the shifts of sections relative to one another  $(\alpha)$ , the duration of hiatuses  $(\delta)$ , the degree of smoothing of the spline  $(\lambda)$ , the extent to which overlap of signal points should be favoured  $(C_{overlap})$ , and optionally the residual standard deviation of the spline  $(\sigma)$ . Although the choice of any of those parameters has the potential to affect posterior alignments and age models, they also offer a chance to explicitly include geological information that could otherwise only be incorporated by discarding or modifying alignment solutions after the algorithmic alignment.

While it is relatively straightforward to express prior beliefs on the alignment parameters  $\alpha$ ,  $\nu$ ,  $\zeta$  and  $\delta$ , it is hard to specify suitable priors for  $\lambda$ ,  $\sigma$  and  $C_{overlap}$ , as they do not correspond to measures used by geologists. The default priors on  $\lambda$ ,  $\sigma$  and  $C_{overlap}$  in the StratoBayes software were chosen iteratively by working with various test data sets. Users should avoid fine-tuning these priors directly on the data sets to which they intend to apply StratoBayes, as this could introduce unintended circularity. Instead, analogous independent data sets could be used to identify suitable priors for  $\lambda$ ,  $\sigma$  and  $C_{overlap}$ . For example, priors on  $\lambda$  and  $\sigma$  for correlating  $\delta^{13}$ C curves could be meaningfully specified from pre-existing reconstructed  $\delta^{13}$ C composite curves.

#### 5.2.3 Challenges with the proxy and sedimentary record

Chemostratigraphy, and, more broadly, correlating geological sections based on proxy data relies on the proxies accurately reflecting a common, underlying signal. Several processes may disrupt this assumption. For example,  $\delta^{13}$ C recorded in carbonates differs between different depositional environments, water depths, and grain types (Geyman and Maloof, 2021), while the  $\delta^{13}$ C recorded in restricted basins may be offset significantly relative to contemporary carbonates elsewhere (Uhlein et al., 2019). Where known, such offsets could be accounted for by subtracting or adding the estimated offset relative to global values. Alternatively, anticipated offsets could be modelled as additional unknown variables, as in Edmonsond and Dyer (2024). This approach will likely require substantial prior knowledge on the potential magnitude and direction of offsets; otherwise, the combination of variation along the height or time axis and along the proxy value axis may result in a large range of mathematically feasible alignments.

A more fundamental problem is posed when similar patterns in a proxy curve are asynchronous in different sections: Shifting and stretching proxy data from multiple sites may result in a strongly correlated composite curve, but this correlation does not prove that the patterns or excursions observed at different sites were in fact synchronous (Blaauw, 2012). Unless supported by independent evidence such as precise radiometric dates, relative age estimates derived from proxy correlations (e.g.  $\delta^{13}$ C) are conditional on the assumption of synchronicity.

Several challenges arise from the variability of sedimentation and the incompleteness of the sedimentary record. Sediment accumulation rates vary with measurement scale (Sadler, 1981): closer spacing between measurements allows more variability to be identified, with actual sedimentation rate histories displaying fractal properties (Miall, 2015). This implies that depositional ages tend to vary non-linearly along a vertically sampled sedimentary section, with substantial incompleteness in shallow-water records (Curtis et al., 2025). These discontinuities can lead to drastically altered shapes of proxy curves from different depositional settings, and cycles from periodic proxy fluctuations may be missed due to insufficient

preservation or sampling (Curtis et al., 2025). This issue is evident in the Sukharikha section, where it is somewhat ambiguous whether the hiatus represents a fraction of a  $\delta^{13}$ C excursion (alignment 1 and 2) or extends over more than one full cycle (alignment 3, Fig. 6). For correlations within sedimentary basins, the method of Bloem and Curtis (2024) could help resolve ambiguous alignments by reconstructing depositional histories through geological process modelling, but this method requires exceptionally high-resolution sampling and its utility has yet to be demonstrated with real-world data sets.

Besides the completeness, the sampling density of proxy records may influence correlations. In StratoBayes, densely sampled sections or parts of sections exert more influence on the shape of the spline than those that are thinly sampled, which can be seen in the spline curve primarily following the densely sampled Oued Sdas and Talat n'Yissi records in Fig. 6. Despite this, our Cambrian case study demonstrates that sections with differing sampling densities – both between and within sites – can still be effectively aligned. Varying sampling density would, however, pose a challenge for reconstructing a global average proxy curve from local records, as the global curve would primarily reflect the more densely sampled sites.

StratoBayes introduces a simplification in modelling sedimentary histories by forcing uniform sedimentation rates within pre-defined segments of a stratigraphic section. An effect of this simplification can be seen in the age-depth plots in Fig. 7: Due to sedimentation rates being modelled as uniform within stratigraphic partitions, the uncertainty of age estimates does not necessarily increase away from the radiometric dates. We acknowledge that this may underestimate the uncertainty associated with potential sedimentation rate variability (De Vleeschouwer and Parnell, 2014), especially when allowing for few sedimentation rate changes. Similarly, our method currently only allows for specifying potential hiatuses with an unknown duration at fixed, predetermined heights.

In principle, our method could be used to divide stratigraphic sections into an arbitrary number of segments with differing sedimentation rates, and with an arbitrary number of potential hiatuses. In practice, estimating the parameters of a model with more than a low double-digit number of alignment parameters (shift parameters, sedimentation rates, hiatuses) represents a challenge for the current implementation of the MCMC algorithm within StratoBayes, as finding and exploring the posterior becomes increasingly difficult as more parameters are added. This limitation could be alleviated by incorporating MCMC methods suited for higher dimensional problems and difficult posterior geometries. Alternatively, a continuous process model such as the compound Poisson-gamma process of BChron (Haslett and Parnell, 2008) might be integrated with our model for the proxy signal, but again the complexity of the MCMC would increase. Another approach would be to divide the alignment problem into sub-problems, e.g. by multiple pairwise correlation of sites (e.g. Hagen et al., 2024; Sylvester, 2023), or by correlating shorter subsections.

## 5.3 Towards quantitative stratigraphy

- Quantitative stratigraphic correlation and age modelling of diverse geological data represent a long-standing challenge in stratigraphic research. Although many algorithms exist for correlating geochemical and geophysical stratigraphic data (e.g. Baville et al., 2022; Bloem and Curtis, 2024; Hay et al., 2019; Sylvester, 2023); few can readily provide uncertainty estimates or incorporate different types of data simultaneously (e.g. Al Ibrahim, 2022; Edmonsond and Dyer, 2024; Lee et al., 2022). Consequently, integrated statistical approaches have only rarely been applied to complex real-world stratigraphic problems (Hagen et al., 2024; Lee et al., 2022).
- Our new method has the potential to be applied to diverse datasets; examples range from shallow borehole data from the Holocene (Finlay et al., 2022) to Proterozoic carbonates (Halverson et al., 2010). The ability of our model to incorporate multiple proxy records simultaneously opens new possibilities for refining stratigraphic correlations. For instance, correlations involving both  $\delta^{13}$ C and  $\delta^{87}$ Sr records could benefit from a probabilistic framework that accounts for their respective uncertainties (Bowyer et al., 2022). The integration of multiple proxies, e.g. multiple element ratios, in the StratoBayes framework could allow correlations based on the entire record of all proxies, rather than a few visually distinct transitions (Craigie, 2015).
- Beyond geochemical records, our approach could also be applied e.g. to geophysical well-logs such as gamma ray or density logs, and magnetostratigraphic records could be correlated directly rather than relying on visually interpreted polarity reversals (Langereis et al., 2010). While index fossils can currently be integrated as tie points, the modelling framework could be expanded to explicitly model first and last occurrences to better incorporate biostratigraphic uncertainty. Similarly, astrochronological constraints can be expressed as priors on sedimentation rates, but an additional model component would be needed to incorporate all astrochronological information from a given site (Sinnesael et al., 2024).

## **Conclusions**

StratoBayes is a Bayesian modelling framework for the probabilistic alignment of stratigraphic proxy records and age modelling. It correlates quantitative proxy signals such as isotope ratios, and integrates additional stratigraphic information such radiometric dates, to construct probabilistic age models. Applying our model to both simulated data and real-world stratigraphic records from the lower Cambrian of Morocco and Siberia, we have demonstrated its ability to account for uncertainty from all model components and to identify multiple plausible alignment solutions. Our lower Cambrian case study provides a fully

- probabilistic estimate for the first appearance of trilobites in Siberia, and quantifies the temporal gap
- between their first occurrence and the oldest Moroccan trilobites. While our results remain dependent on
- model assumptions, they represent a step towards a more objective and reproducible approach to early
- Palaeozoic stratigraphy; they also highlight sources of uncertainty and identify targets for future research.
- Beyond this case study, StratoBayes has broad applicability to stratigraphic problems across all time
- intervals that involve the correlation of quantitative proxy records.

## Appendix A: Markov chain Monte Carlo sampling scheme

- Appendix A details the Metropolis-within-Gibbs sampling scheme and the parallel tempering framework
- that are used within the StratoBayes software to sample from the posterior of the unknown model
- parameters.

721

740

## Sampling strategy

- The MCMC sampling scheme used in this study includes an adaptive phase. During this phase, proposal
- distributions and the probabilities with which different proposal types are selected for the Metropolis-
- Hastings updates are adjusted based on the history of the MCMC chains to improve acceptance rates and
- mixing. Additionally, the temperature ladder of the parallel tempering framework is updated to improve the
- swap rates of chains. After the adaptive phase, the proposal distributions and probabilities, as well as chain
- temperatures, remain fixed for the remainder of the run to ensure proper sampling from the posterior.
- In the current implementation, the length of the adaptive phase is pre-determined by the user, specified as
- a fixed number of iterations. However, the user has the option to extend the adaptation period by continuing
- the run if needed. More generally, adaptation could also be stopped automatically based on criteria such as
- mixing within chains (Yang and Rosenthal, 2017) or convergence criteria.
- Adaptive MCMC algorithms do not always preserve the stationarity of the target distribution during the
- adaptive phase (Roberts and Rosenthal, 2009). Therefore, all samples from the adaptive phase are discarded
- as burn-in. Additionally, if diagnostic checks suggest that the MCMC has not converged by the end of the
- adaptive phase, further samples may need to be discarded.

## Gibbs sampling scheme for the cubic B-splines

- The following sampling scheme was adapted from Heaton et al. (2020). The spline coefficients are sampled
- from a multivariate normal distribution of the form:

$$\beta \sim MVN(b\mathbf{Q},\mathbf{Q}), \qquad (16)$$

where b is given by .

$$b = (\mathbf{B}(h))^T \frac{y}{\sigma^2}, \qquad (17)$$

- $\mathbf{B}(h)$  are cubic B-splines (Eilers and Marx, 1996) at a set of k knots evaluated at heights h at which y, the
- composite stratigraphic signal of all sites, was observed. Here,  $\sigma$  is the residual standard deviation.
- The other element needed for sampling from the posterior of b is Q, given by

$$\mathbf{Q} = (\mathbf{H} + \lambda \mathbf{D})^{-1}, \qquad (18)$$

where  $\lambda$  is a smoothing parameter, **D** is a penalty matrix to prevent the spline from overfitting the data, and

$$\mathbf{H} = \left(\frac{\mathbf{B}(h)}{\sigma}\right)^T \frac{\mathbf{B}(h)}{\sigma} \tag{19}$$

The standard deviation  $\sigma$  can be fixed as

$$\sigma = \frac{1}{S} \sum_{s=1}^{S} \sigma_s, \qquad (20)$$

- where S is the number of sites, and  $\sigma_S$  is the standard deviation of individual splines fitted to the data of site
- 755 s. This often provides a good approximation of  $\sigma$ , while removing an unknown model parameter, potentially
- facilitating quicker convergence of the model run.
- Alternatively,  $\sigma$  can be estimated within the Gibbs sampling scheme from the data, by placing a conjugate
- gamma prior on the inverse of the variance (precision,  $\tau = 1/\sigma^2$ ):

$$\sigma^{-2} \sim Gamma \left( a_{\sigma} + \frac{n_{y}}{2}, b_{\sigma} + \frac{1}{2} \sum_{\alpha} \left( y - \beta \mathbf{B}(h) \right)^{2} \right)$$
 (21)

The smoothing parameter  $\lambda$  is estimated by placing a gamma prior on  $\lambda$ :

$$\lambda \sim Gamma \left( a_{\lambda} + \frac{k}{2}, \frac{1}{\frac{1}{b_{\lambda}} + \frac{1}{2} \sum^{k} \beta \mathbf{D} \times \beta} \right)$$
 (22)

#### **Metropolis-Hastings step**

- The starting heights or ages  $\alpha$ , sedimentation rates  $\nu$ , site multipliers  $\zeta$  and gaps  $\delta$  are updated in a
- Metropolis-Hastings step. For each unknown parameter, a new value is randomly sampled from a proposal
- distribution. Initially, proposals are sampled independently for each parameter from its respective prior, or
- alternatively from a custom proposal distribution.
- In the following, the current set of parameter values is labelled  $\theta$ , and the proposed set is labelled  $\theta'$ . To
- decide whether to accept or reject the new set of parameters, an acceptance probability A is calculated, and
- the proposal is randomly accepted or rejected with a probability of A. This probability is calculated as

$$A = min\left(1, \frac{\pi(\theta')}{\pi(\theta)}\right), \qquad (23)$$

- where  $\pi(\theta)$  is the unnormalised posterior probability of the current values, and  $\pi(\theta')$  is the unnormalised
- posterior probability of the proposed values. These can be calculated as

$$\pi(\theta) = p(\theta) \times L(data|\theta), \qquad (24)$$

- where  $p(\theta)$  is the prior probability of  $\theta$ , and  $L(data|\theta)$  the likelihood of the data given  $\theta$ .
- We calculate the likelihood of the data given  $\theta$  as a product of the probability densities of each data point
- of the signal y (recorded at two or more sites) and of all absolute age information. For the signal, we assume
- that the observed values y are normally distributed and centred around the values predicted by the splines,
- $\mu$ , at height h, with a standard deviation  $\sigma$  which has been introduced earlier. The likelihood of a data point
- i from the signal y is thus

$$L(y_i|\theta) = \frac{1}{\sqrt{2\pi\sigma^2}} \times e^{\left(-\frac{(y_i - \mu_i)^2}{2\sigma^2}\right)}$$
 (25)

and the log-likelihood for all data points of the signal is calculated as

$$ln L(y|\theta) = \sum_{i} ln L(y_i|\theta)$$
(26)

- If more than one type of signal is used, the log-likelihood of additional signals can be calculated analogously
- and added in Equation 29.

Age constraints are incorporated by using an age estimate from radiometric dates d with, for example, mean ages  $a_{mean}$  and uncertainties given by standard deviations  $a_{sd}$ . The probability density of a date  $d_i$  is then calculated as

$$L(d_i|\theta) = \frac{1}{\sqrt{2\pi a_{sd,i}^2}} \times e^{\left(-\frac{a_{mean,i} - a_{predicted,i}}{2a_{sd,i}^2}\right)}$$
(27)

- where  $a_{predicted,i}$  is the age predicted by the age-height transform at the height  $h_{s,i}$ , the height at the site at which date  $d_i$  was obtained.
- The log-likelihood for all age constraints is calculated as

$$\ln L(d|\theta) = \sum_{i} \ln L(d_i|\theta) \qquad (28)$$

and the overall likelihood, if absolute age constraints are included, is

$$lnL(y,d|\theta) = lnL(y|\theta) + lnL(d|\theta)$$
 (29)

#### Proposal types

805806

- In order to allow for a broad search of the parameter space, proposals are initially selected independently for each parameter, and are selected independently of the current parameter values. These proposals lead to a decreasing acceptance rate over time, and the chain tends to arrive at a single set of values with high posterior probability,  $\pi(\theta)$ , remaining there for many iterations due to frequent rejections. Therefore, different types of proposals are used after an initial period:
- Proposing from the prior or a custom distribution: This proposal is used exclusively for a small number of initial iterations and is alternated with other proposals later on.
  - 2) Adaptive independent (univariate) proposals: Proposals for each parameter are selected independently from other parameter values. Proposals are dependent on the current state of the parameter  $\theta_i$ , and sampled from a normal distribution  $N(\theta_i, \sigma_i)$ , where  $\sigma_i$  is a standard deviation that is estimated based on the history of the MCMC chain, i.e. based on the sampled  $\theta_i$  from previous iterations.
  - Adaptive dependent (multivariate) proposals (Roberts and Rosenthal, 2009): Proposals for the parameters are selected jointly and are dependent on the current state of the parameters  $\theta$ . Proposals are sampled from a multivariate normal distribution  $MVN(\theta, \Sigma)$ , where  $\Sigma$  is a

- covariance matrix that is estimated based on the history of the MCMC chain, i.e. based on the 812 sampled  $\theta_i$  from previous iterations.
  - 4) Shifting some or all  $\alpha$  and or  $\delta$  parameters while keeping the other parameters constant. This can accelerate the convergence of the MCMC in cases where some sites are aligned with each other, but offset relative to other sites.

Proposal types are chosen with a probability that broadly corresponds to the relative acceptance probability of the respective proposal type, i.e. proposal types that are rejected often are chosen less frequently. Adaptation for types 2) and 3), and the adjustment of proposal type probabilities ends after the adaptive phase. Posterior samples from the adaptive phase have to be discarded as burn-in, to ensure the correct convergence of the chain.

#### Parallel tempering

To avoid the MCMC chain becoming trapped at isolated peaks of the posterior probability distribution, we implement a parallel tempering framework, following Sambridge (2014). This involves running multiple chains in parallel. The target chain, the chain from which the posterior samples will be taken, is left unaltered ("cold chain"). The other chains are tempered, i.e. their unnormalised log posterior probabilities are raised to the power of 1/T, with T being the temperature. The higher T, the more "flattened" the posterior probability landscape becomes, and the easier it is for the chain to explore the landscape. Frequently, chain swaps are proposed, during which the model parameter values of different chains are exchanged with a Metropolis-Hastings acceptance probability based on the ratios of posterior probabilities of the states of the two chains, evaluated at both temperatures as in Appendix A2 of Sambridge (2014).

The initial temperatures for a number of chains  $n_c$  are selected using a geometric spacing, with  $T_1 = 1$  (cold chain) and  $T_{n_c} = \infty$  (hottest chain). The infinite temperature of the hottest chain implies that all proposals during the MCMC will be accepted, and we let that chain sample from the prior probability distributions of the parameters. If  $n_c > 2$ , intermediate chain temperatures are selected as

$$T_c = 10^{\sum_{j=2}^{c} d_j}, \qquad (30)$$

where

$$d_c = \frac{(n_c - 1)^{(2/3)}}{n_c - 2} + \frac{c - 1 - (n_c - 1)/2}{1.5 * n_c}, \quad c = 2...n_c - 1$$
 (31)

This leads to the spacing of temperatures decreasing with increasing number of chains, and temperature spacing is narrower for lower temperatures on the log scale. A small amount of white noise from a normal distribution with zero mean and a standard deviation of  $(5 \times n_c)^{-1}$  is added to each  $d_c$  to vary the initial temperature ladders between independent model runs. Temperatures are updated in the adaptive phase of the MCMC to increase the swap rates of chains (Vousden et al., 2016).

# **Appendix B: Inspecting the posterior of the lower Cambrian case**

## **study**

- Appendix B provides additional details on the posterior of the inference with lower Cambrian  $\delta^{13}C$  data and
- radiometric dates.

#### Trace plots

Trace plots visualise the evolution of chains from an MCMC and, together with tools such as the potential scale reduction factor (Gelman and Rubin, 1992; Vats and Knudson, 2021), allow for assessing convergence of model runs. The trace plot indicative of a well-behaved model run should be stationary after the burn-in phase, with different chains mixing well (Gelman et al., 1995). An example of a well-behaved trace plot is the first panel of Fig. B1. Inspecting the trace plots of the 18 model parameters of the lower Cambrian case study reveals that all parameters seem to have reached stationarity, this said; some chains occasionally visit distinctly different values (e.g. Fig. B1, column 1, row 2). The chains are not mixing well in those regions of the parameter space. Running the model for considerably more iterations is likely to overcome this problem. However, this affects only the less likely alignments; the most likely alignment (alignment cluster 1) is well explored across all parameters.

Figure B1: Trace plots of the 18 alignment parameters. Each colour corresponds to a distinct run. For visual clarity, only 250 samples are displayed per run. The burn-in phase (the first 150,000 iterations) is omitted.

# Age-depth models for different alignments

The age-depth models for each of the four sites are shown for each alignment cluster separately in Fig. B2 (instead of for all samples combined as in Fig. 7).

Figure B2: Age-depth model for each of the four sites. The solid lines indicate the median posterior ages corresponding to the respective heights; the shaded interval denotes the 95% credible interval of posterior ages. Colours correspond to the three different alignment clusters and outlier samples. Circles indicate the mean age estimates of radiometric dates, with vertical lines spanning two standard deviations around the mean of these age estimates. Crosses denote the first appearances of trilobites in Morocco and Siberia.

# Variation within alignment clusters

Summarising the posterior by grouping samples into clusters of similar alignments facilitates discussion of the results but risks oversimplifying the variation within each cluster. Each cluster represents a set of posterior samples that share similar inferred ages for the partition boundaries, but differences still exist between individual samples within the same cluster. As an example, three distinct alignments from cluster 1 are visualised in Fig. B3. An alignment from a sample not assigned to any cluster is shown in Fig. B3d.

Figure B3: Alternative alignments, each corresponding to a single sample from the posterior. (a) A sample from the most likely cluster 1, corresponding to that shown in Fig. 6a. (b, c) Alignments corresponding to other samples from cluster 1. (d) Alignment corresponding to an outlier sample that was not assigned to any cluster. The curved dark lines show the cubic B-splines corresponding to each alignment.

## Posterior of alignment parameters

The posterior distributions of the alignment parameters are summarised in histograms in Fig. B4.

Figure B4: Comparison of prior and posterior probability densities. Histograms in colour denote the posterior probability densities of the 18 alignment parameters; the grey, smooth shadings represent prior probability densities. The four colours correspond to the four independent model runs.

## Code and data availability

The StratoBayes R package is available for download as a binary at stratobayes.github.io. The data and R scripts used to generate the results are available at https://zenodo.org/records/15065336.

#### **Author contribution**

MRS and ARM conceived the idea. MRS, ARM and MS designed the initial model version, and all authors contributed to the design of the final model. MS, ARM and MRS coded the software prototype. KE, MRS and ARM modified and expanded the software. MS curated the lower Cambrian data. KE conducted the analyses and visualised the results. KE wrote the initial manuscript draft, and all authors contributed to the final version. MRS coordinated the project. MRS and ARM acquired the funding.

### **Competing interests**

883

885

The authors declare that they have no conflict of interest.

## Acknowledgements

- We thank the participants of the StratoBayes workshop (Durham, 2024) for feedback on an early version
- of the StratoBayes software. We thank Tim Heaton for sharing his code for spline fitting. We thank Andrew
- Valentine for discussions on the model. We thank Prof. Nasrrddine Youbi and Kamal Mghazli for hosting
- an exploratory field visit to the lower Cambrian sections in Morocco. We thank Maarten Blaauw, Andrew
- Curtis and Norman MacLeod for their helpful feedback on an earlier version of this manuscript.

## Financial support

This research was funded by Leverhulme Research Project Grant 2019-223 to MRS.

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
