# Peer review of "StratoBayes: A Bayesian method for automated stratigraphic correlation and age modelling"

_EGUsphere, 2025_

## Referee Comment (RC2)

**EGU Sphere, Eichenseer et al., (2025):**

*"StratoBayes: A Bayesian method for automated stratigraphic correlation and age modelling"*

Review by Andrew Curtis, University of Edinburgh

This manuscript proposes a Monte Carlo based method to assess quantitative Bayesian uncertainties in the statigraphic correlation between data transects recorded at different locations. It applies the method to both synthetic and real data, to highlight strengths and weaknesses of the method.

Overall I feel that this work will make a significant advance over manual correlation methods and some other algorithmic methods. I nevertheless have a few comments which all concern the methodology rather than the applications.

**Main comments:**

1. In principle, a hiatus may occur between any two measurement locations in a geochemical transect (Sadler, 1981), as the authors acknowledge in the discussion at the top of page 29. For example, hiatuses certainly occur in age records at grain scale, in any grainy depositional setting. The reason that age correlation still has some validity derives from an assumption that these types of hiatus are insignificant, or can be represented in aggregate as smooth increases in age, when averaged over the time scales at which data sets are correlated.

Nevertheless, when we try to assess uncertainty in correlations, any possibility that hiatuses exist which are significant even at these longer time scales should produce a variation in the resulting age curves along the measurement axis or transect, and so should be considered in uncertainty estimates. In a Bayesian context, information (beliefs or constraints) on inter-datum hiatuses should therefore ideally be expressed explicitly as so-called prior information, embodied in prior probability distributions. Such prior information may come from, for example, the fact that at some scale of observation we have / have not observed a sedimentological sign of hiatus in the rock record along the transect or in neighbouring synchronously deposited sediments, a sedimentary process or statistical model that embodies such beliefs, or any other type of pertinent observation. In the current algorithm, the uncertainty in these sources of information is not included – the locations of hiatuses are defined definitively *a priori*. The authors do acknowledge the possibility that hiatuses may occur anywhere on page 29, but they leave that problem unsolved.

I feel that it is important to make this prior probability distribution (the *probability* that a hiatus exists at any point in the transect) explicit in, what the authors propose is, a general correlation method: this will force practitioners to consider properly the *quality* of information, and the *degree* of expert belief or received opinion, that are fed into the process. Ideally all of the number, locations/age and length of hiatuses would be subject to uncertainties, and practitioners would therefore be forced to decide not only whether/where they can definitively fix a hiatus, but also where they might have missed one or more, and how likely that was to have happened.

I would also admit that translating geological observations into prior probabilities that can be incorporated into the Bayesian inference is not at all straightforward! It requires a process of expert elicitation, which is itself a significant challenge and source of epistemic uncertainty. Polson and Curtis (2010), Bond et al., (2012), Curtis (2012) and Bond (2015) describe some of the biases that affect geological interpretations by experts, and techniques that might be adopted to minimise

these. While these papers are concerned with different types of observations, similar types of biases and uncertainties can be expected in the cases described above – perhaps in addition to other biases that are particular to those cases.

2. This method, along with most others employed for stratigraphic correlation, contains the implicit assumption that signals in the data from each synchronous sedimentary package exhibit similar data values. However, this is clearly not always the case, for example when comparing records in shallow and deeper water settings, or any other settings in which one record is more prone to have missing sections than another. This is discussed by the authors at the top of p.29. One approach to address this is to incorporate explicit relationships between the patterns that one might expect in different contemporaneous settings, such as might be embodied in a geological process model (e.g., Bloem et al., 2024); the authors rightly say that the latter approach has not been tested on real data, but then leave this problem hanging, with no other suggested solution.

Yet this effect may not be minor, and similar to comment 1 above, might introduce significant epistemic uncertainty that is not currently accounted for in the authors' algorithm. If the authors aim to assess uncertainty then ideally they would think about how to embody, or at least make explicit, all sources of uncertainty that affect age correlations, so I suggest that some further discussion would be valuable, about how one might address this in future.

3. In a number of previous studies, each geochemical proxy data set is often scaled in magnitude, in order to better match one transect to another. Was this not done in the current study (or did I miss it)? Differences in depositional environment, as mentioned in comment 2, may result in signals having different magnitudes, so is there a need to include such a scaling (perhaps *a priori*) to match the magnitudes of signals from one transect to another?

4. There is almost no way that a geologist can assess suitable values, or even ranges of values, for some of the parameters employed in the authors' algorithm (e.g., lambda) *a priori* – without looking at *any* outputs of the process). It is therefore not possible to define the corresponding prior information. I would guess that anyone applying this algorithm will use a trial and error approach to vary such parameters, running the algorithm each time, looking at the results until they get a good result, where 'good' then becomes entirely subjective. The parameter is then in fact defined *a posteriori* in a pseudo-hierarchical way – but without ever defining its prior distribution. I think that an example of this is even given by the authors themselves, in lines 281 to 285. As a result, while this method looks Bayesian mathematically, in practical use I fear that it might not be.

How can the authors change or differently embody these parameters to provide geologists with an intuitive way to define them *a priori*? One possibility might be to use the inverse approach to define prior information from Curtis and Wood (2004) or Walker and Curtis (2014), but are there others? Generally, it seems to me that if this methodology is to make a significant impact, in making the quantification of uncertainty in correlations more objective, then more research (perhaps in other papers) and discussion (in this paper) is needed to develop structured methods to define the prior distributions; otherwise this method may well be used in a similarly subjective manner to manual correlation, and while the results will be quantitative, they could end up being little more objective and well defined than previous results.

5. Lines 228-232 indicate that uncertainty may not increase with distance from absolute age constraints. I agree with the previous reviewer that this seems to indicate a significant flaw in the methodology. It may

again be due to the particular implementation – perhaps the density of spline knots should increase with distance away from the absolute age constraints (although it is not clear how quickly).

A similar issue arises close to sequence boundaries, around which time tends to be compressed in the stratigraphic record. Spline knots might be more densely distributed around such boundaries, but again it is not clear how dense they should be. This is another case where defining prior information is difficult, and requires more study (similar to the comment above).

**Minor Comments:**

Line 130-131: This sentence needs some explanation; the main text should be understandable without having to read the Appendices.

Fig. 1 caption: as far as I can tell, both alpha and gamma are used in the main text before they are defined, other than in this figure caption.

**REFERENCES**

C. E. Bond. Uncertainty in structural interpretation: Lessons to be learnt, Journal of Structural Geology, 74, 2015, https://doi.org/10.1016/j.jsg.2015.03.003

C. E. Bond, R.J. Lunn, Z.K. Shipton, A.D. Lunn; What makes an expert effective at interpreting seismic images?. Geology 2012;; 40 (1): 75–78. doi: https://doi.org/10.1130/G32375.1

A. Curtis, 2012. The science of subjectivity. Geology. 40, pp. 95-96. doi: 10.1130/focus012012.1

Andrew Curtis, Hugo Bloem, Rachel Wood, Fred Toby Bowyer, Graham Anthony Shields, Ying Zhou, Mariana Yilales, Daniel Tetzlaff, 2025. Natural sampling and aliasing of marine geochemical signals. Scientific Reports, 15:760, DOI: 10.1038/s41598-024-84871-6

A. Curtis and R. Wood 2004. Optimal elicitation of probabilistic information from experts. In, Geological Prior Information, A. Curtis and R. Wood ed's. Geol. Soc. Lond. Special Publication, Vol. 239; pp. 127-145; DOI 10.1144/GSL.SP.2004.239.01.09

D. Polson and A. Curtis, 2010. Dynamics of uncertainty in geological interpretation. Journal of the Geological Society, London, Vol. 167, pp. 5-10. doi: 10.1144/0016-76492009-055

M. Walker and A. Curtis, 2014. Expert elicitation of geological spatial statistics using genetic algorithms. Geophys. J. Int., 198, pp.342–356, doi: 10.1093/gji/ggu132

---

## Author Comment (AC1)

**Response to the comments of Reviewer 1 (Maarten Blaauw)**

We thank the reviewer for his time and helpful feedback. Our responses to the reviewer's comments (black) are recorded below in blue.

**Reviewer 1**

This manuscript proposes a new method to align multiple proxy records based on assumed synchroneity (e.g. appearance of key trilobite fossils); additional data such as radiometric dates or known ages of fossils can also be added. The model draws a Bayesian cubic spline (Heaton et al., 2020) per to-be-aligned proxy, using evenly-spaced knots and smoothness parameters. The model is applied to some synthetic and real-world examples.

I like the fact that not just one alignment is chosen, displayed and discussed, but a range of alignments (e.g., Fig. 6 and section 5.1). This clearly shows the probabilistic and uncertain nature of aligning multiple records, and thus the need and potential for a Bayesian framework. Could the age-depth relationships of the three solutions from Fig. 6 also be shown in a Figure akin to Fig. 7, to see how variable the reconstructed rates and hiatuses are?

We thank the reviewer for this positive assessment and have included an additional figure in Appendix B to show the age-depth relationships of the three different solutions from Fig. 6. (i.e. Fig. B2).

Sometimes stratigraphical correlation is the only way to obtain a chronology for a proxy record, e.g. where no absolute/radiometric age estimates are available. However, it would be good to also highlight potential problems with aligning records based on their assumed synchroneity, e.g. problems with circular reasoning, possible erroneous choice of tie-points, and the introduction of a dependence between records. These problems are reviewed by Blaauw 2012 (doi:10.1016/j.quascirev.2010.11.012).

We expanded the discussion in section 5.2.3 to point out those challenges:

*A more fundamental problem is posed when similar patterns in a proxy curve are asynchronous in different sections: Shifting and stretching proxy data from multiple sites may result in a strongly correlated composite curve, but this correlation does not prove that the patterns or excursions observed at different sites were in fact synchronous (Blaauw 2012). Unless supported by independent evidence such as precise radiometric dates, relative age estimates derived from proxy correlations (e.g. $\delta^{13}C$) are conditional on the assumption of synchronicity.*

Line 76, would it be useful to mention Trayler et al. 2024's Astrobayes age-model, which includes hiatuses (doi:10.5194/gchron-6-107-2024)?

*Reference added in lines 68-70: A Bayesian age-depth modelling approach by Trayler et al. (2024) considers hiatuses and uses astrochronological interpretations to inform sedimentation rate priors.*

Lines 228-32 and 646-52 list an important limitation of the proposed model; assumed linear sedimentation rates will not cause chronological uncertainties to widen further away from age constraints. Some of the reconstructed age-model uncertainties seem very narrow indeed, e.g. 7d. Does setting spline knots at regular intervals not help?

*The age model uncertainties at 7d (Siberia) are low because of the high-amplitude variations in $\delta^{13}C$ which are matched with similar high-amplitude signals at Oued Sdas, leading to comparatively low uncertainties. Spline knots are placed at regular intervals, but for age uncertainties to widen away from age constraints, additional sedimentation rate changes would have to be included (see discussion in section 5.2.3, lines 672 – 690).*

For a frequently-used Bayesian age-depth model that includes priors on sedimentation rates and variability, please cite Bacon (Blaauw & Christen 2011, https://projecteuclid.org/euclid.ba/1339616472). Bacon is a piece-wise linear model much like what is proposed here; it also includes time hiatuses, slumps (depth 'hiatuses') and changes in sedimentation rates. It uses the t-walk, a flexible MCMC (Christen & Fox 2010, http://projecteuclid.org/euclid.ba/1340218339). Although Bacon is most often used on radiocarbon-dating timescales, it has also been applied to much longer time-scales. That said, the usage of dozens of parameters per site (owing to long cores with thin sections) would probably cause the MCMC to run much, much slower than the 5 days reported here.

*We thank the reviewer for pointing out this oversight and have expanded the introduction to refer to Bacon and other Bayesian age-depth modelling tools (lines 65 – 68): Bayesian approaches are commonly employed in age-depth models that interpolate between absolute age constraints or tie points, e.g. Bchron (Haslett and Parnell, 2008) and Oxcal (Ramsey 1995); Bacon (Blaauw and Christen 2011) also includes priors on sedimentation rates and variability.*

I ran a quick toy age-model in R using the vignette provided and all ran fine. This is important, because other recently proposed methods I've seen rely on many additional packages and on software external to R such as JAGS to run (often resulting in failure). Pity though that only binary versions are provided - could the source c++ code also be provided? That would enable users on other operating platforms to also run the code, would enable users to get a better idea of what exactly is done, and would be much more future-proof.

We are glad that the reviewer found that the package ran smoothly. We agree that releasing the source code would be beneficial, and plan to do so once we have more clarity on the long-term direction of this software project.

The MCMC runs multiple chains but only retains the samples from one chain (both a burn-in and thinning are applied afterward). Is this a standard approach?

The discarded chains are tempered chains; we are using a parallel tempering framework for easier sampling of multimodal posteriors. We have expanded the explanation in lines 136 – 139 to clarify this:

*To ensure thorough exploration of the parameter space, we employ parallel tempering, i.e. we run multiple chains in parallel, flattening the likelihood of the tempered (hot) chains which can therefore move between different posterior modes, and frequently propose swaps between chains. For the posterior estimates, we retain samples only from the primary (cold) chain.*

Could you clarify $\mu$ in section 2.1: is this a hypothetical target to which all sites are tuned, or is this akin to target/reference Site 1 as in Fig. 1?

μ in section 2.1 is a hypothetical target, to which all sites except the reference site are tuned (the reference site remains fixed in this scenario, but is informing μ alongside the tuned sites). We have added a clarification in section 2.1 (avoiding the term "target" and referring to a hypothetical composite curve instead): *Here, μ can be interpreted as an underlying common signal of which the observations from each site, including the reference site, are noisy realisations.* (lines 154 – 155)

Fig. 1 of the hypothetical sample: can the $\alpha$ and $\gamma$ values of the placement in c) be depicted as vertical lines overlying the prior distributions of panel b)? This because in this example, site 2 is compressed a lot (2.8 times faster than site 1), and it would be nice to see where it falls on the log-normal prior (as well as of course the placement on the uniform prior, 12.5 m). In this example, site 2 accumulates linearly over time.

This is a helpful improvement of Fig. 1, we've added the reviewer's suggestion to the updated version of Fig. 1.

Eq. 8, shouldn't the hiatuses $\delta$ be expressed as gaps/jumps in time, not depth/height?

In Eq. 8, correlation is done on a reference height (or reference depth) scale, rather than a time scale, so hiatuses are expressed as an interval of heights on the reference scale that is not represented in the correlated section. A clarification has been added to

section 2.2.3 (lines 216 – 217): *In a correlation on an absolute age scale (Sect. 2.2.5), hiatuses would instead be expressed as durations, not heights.*

**Citation**: https://doi.org/10.5194/egusphere-2025-1355-RC1

---

## Author Comment (AC2)

**Response to the comments of Reviewer 2 (Andrew Curtis)**

We thank the reviewer for his time and helpful feedback. Our responses to the reviewer's comments (black) are recorded below in blue.

**Reviewer 2**

This manuscript proposes a Monte Carlo based method to assess quantitative Bayesian uncertainties in the statigraphic correlation between data transects recorded at different locations. It applies the method to both synthetic and real data, to highlight strengths and weaknesses of the method. Overall I feel that this work will make a significant advance over manual correlation methods and some other algorithmic methods. I nevertheless have a few comments which all concern the methodology rather than the applications.

**Main comments:**

1. In principle, a hiatus may occur between any two measurement locations in a geochemical transect (Sadler, 1981), as the authors acknowledge in the discussion at the top of page 29. For example, hiatuses certainly occur in age records at grain scale, in any grainy depositional setting. The reason that age correlation still has some validity derives from an assumption that these types of hiatus are insignificant, or can be represented in aggregate as smooth increases in age, when averaged over the time scales at which data sets are correlated.

Nevertheless, when we try to assess uncertainty in correlations, any possibility that hiatuses exist which are significant even at these longer time scales should produce a variation in the resulting age curves along the measurement axis or transect, and so should be considered in uncertainty estimates. In a Bayesian context, information (beliefs or constraints) on inter-datum hiatuses should therefore ideally be expressed explicitly as so-called prior information, embodied in prior probability distributions. Such prior information may come from, for example, the fact that at some scale of observation we have / have not observed a sedimentological sign of hiatus in the rock record along the transect or in neighbouring synchronously deposited sediments, a sedimentary process or statistical model that embodies such beliefs, or any other type of pertinent observation. In the current algorithm, the uncertainty in these sources of information is not included – the locations of hiatuses are defined definitively a priori. The authors do acknowledge the possibility that hiatuses may occur anywhere on page 29, but they leave that problem unsolved.

I feel that it is important to make this prior probability distribution (the probability that a hiatus exists at any point in the transect) explicit in, what the authors propose is, a general correlation method: this will force practitioners to consider properly the quality of information, and the degree of expert belief or received opinion, that are fed into the

process. Ideally all of the number, locations/age and length of hiatuses would be subject to uncertainties, and practitioners would therefore be forced to decide not only whether/where they can definitively fix a hiatus, but also where they might have missed one or more, and how likely that was to have happened.

I would also admit that translating geological observations into prior probabilities that can be incorporated into the Bayesian inference is not at all straightforward! It requires a process of expert elicitation, which is itself a significant challenge and source of epistemic uncertainty. Polson and Curtis (2010), Bond et al., (2012), Curtis (2012) and Bond (2015) describe some of the biases that affect geological interpretations by experts, and techniques that might be adopted to minimise these. While these papers are concerned with different types of observations, similar types of biases and uncertainties can be expected in the cases described above – perhaps in addition to other biases that are particular to those cases.

We thank the reviewer for the thorough exploration of the problem of hiatuses. We admit that the treatment of hiatuses in the current version of the StratoBayes algorithm is limited: It requires specifying horizons at which hiatuses are suspected to occur a priori. However, as the duration of a hiatus can be expressed as a prior probability distribution, this can allow the algorithm to identify either the duration of the hiatus, or identify that there was no substantial hiatus after all (posterior estimate of the duration of the hiatus close to zero).

We agree that it would be preferable to place a prior distribution on the number of hiatuses, as well as on their position (rather than specifying a fixed number of potential hiatuses at fixed positions). Although extending the model to infer both the number and positions of hiatuses is mathematically feasible, doing so would greatly enlarge the parameter space of the model and demand substantial improvements to the MCMC scheme to achieve convergence within practical run-times. The current version of our method therefore does not allow for including uncertainty about the number and position of hiatuses.

We have slightly expanded the discussion in section 5.2.3 to reference hiatuses (lines 679 – 681):

*Similarly, our method currently only allows for specifying potential hiatuses with an unknown duration at fixed, predetermined heights.*

*In principle, our method could be used to divide stratigraphic sections into an arbitrary number of segments with differing sedimentation rates, and with an arbitrary number of potential hiatuses.*

2. This method, along with most others employed for stratigraphic correlation, contains the implicit assumption that signals in the data from each synchronous sedimentary package exhibit similar data values. However, this is clearly not always the case, for example when comparing records in shallow and deeper water settings, or any other settings in which one record is more prone to have missing sections than another. This is discussed by the authors at the top of p.29. One approach to address this is to incorporate explicit relationships between the patterns that one might expect in different contemporaneous settings, such as might be embodied in a geological process model (e.g., Bloem et al., 2024); the authors rightly say that the latter approach has not been tested on real data, but then leave this problem hanging, with no other suggested solution.

Yet this effect may not be minor, and similar to comment 1 above, might introduce significant epistemic uncertainty that is not currently accounted for in the authors' algorithm. If the authors aim to assess uncertainty then ideally they would think about how to embody, or at least make explicit, all sources of uncertainty that affect age correlations, so I suggest that some further discussion would be valuable, about how one might address this in future.

We agree with the reviewer that offsets in the proxy values or asynchronous proxy patterns interfere with the model assumptions. We point out these issues in section 5.2.3, and have now added a paragraph on the synchronicity assumption following the comments from reviewer 1 (lines 646 – 651):

*A more fundamental problem is posed when similar patterns in a proxy curve are asynchronous in different sections: Shifting and stretching proxy data from multiple sites may result in a strongly correlated composite curve, but this correlation does not prove that the patterns or excursions observed at different sites were in fact synchronous (Blaauw 2012). Unless supported by independent evidence such as precise radiometric dates, relative age estimates derived from proxy correlations (e.g. $\delta^{13}C$) are conditional on the assumption of synchronicity.*

We suggest in section 5.2.3 that the user may account for known proxy offsets by *subtracting or adding the estimated offset relative to global values* (lines 640 – 641). Alternatively, the model could be expanded to model anticipated offsets as additional unknown variables (lines 642 – 643).

3. In a number of previous studies, each geochemical proxy data set is often scaled in magnitude, in order to better match one transect to another. Was this not done in the current study (or did I miss it)? Differences in depositional environment, as mentioned in comment 2, may result in signals having different magnitudes, so is there a need to include such a scaling (perhaps a priori) to match the magnitudes of signals from one transect to another?

In this study, we did not scale the $\delta^{13}C$ data. Whilst scaling may improve the correlation of transects from different sites, this may also introduce spurious fits of asynchronous peaks of a different magnitude. We would caution against scaling data *a priori* unless there is independent evidence for a systematic offset, e.g. due to different depositional environments. We have added a clarification that no scaling was done in lines 360 – 361:

*$\delta^{13}C$ values were used as reported by the authors of the respective publications without any scaling or other adjustments.*

4. There is almost no way that a geologist can assess suitable values, or even ranges of values, for some of the parameters employed in the authors' algorithm (e.g., lambda) a priori – without looking at any outputs of the process). It is therefore not possible to define the corresponding prior information. I would guess that anyone applying this algorithm will use a trial and error approach to vary such parameters, running the algorithm each time, looking at the results until they get a good result, where 'good' then becomes entirely subjective. The parameter is then in fact defined a posteriori in a pseudo-hierarchical way – but without ever defining its prior distribution. I think that an example of this is even given by the authors themselves, in lines 281 to 285. As a result, while this method looks Bayesian mathematically, in practical use I fear that it might not be.

How can the authors change or differently embody these parameters to provide geologists with an intuitive way to define them a priori? One possibility might be to use the inverse approach to define prior information from Curtis and Wood (2004) or Walker and Curtis (2014), but are there others? Generally, it seems to me that if this methodology is to make a significant impact, in making the quantification of uncertainty in correlations more objective, then more research (perhaps in other papers) and discussion (in this paper) is needed to develop structured methods to define the prior distributions; otherwise this method may well be used in a similarly subjective manner to manual correlation, and while the results will be quantitative, they could end up being little more objective and well defined than previous results.

The priors on lambda and sigma, and the overlap prior, are indeed difficult to justify. We agree that future work on how to specify these priors in a more principled way would be valuable. As this is beyond the scope of this manuscript, we have added a word of caution on the potential circularity of tuning priors with the same data that will be used in the analysis, and a suggestion on how suitable priors may be identified in future work in section 5.2.2 (lines 628 – 634):

*While it is relatively straightforward to express prior beliefs on the alignment parameters $\alpha, \gamma, \zeta$ and $\delta$, it is hard to specify suitable priors for $\lambda, \sigma$ and $C_{overlap}$, as they do not correspond to measures used by geologists. The default priors on $\lambda, \sigma$ and $C_{overlap}$ in the*

*StratoBayes software were chosen iteratively by working with various test data sets. Users should avoid fine-tuning these priors directly on the data sets to which they intend to apply StratoBayes, as this could introduce unintended circularity. Instead, analogous independent data sets could be used to identify suitable priors for $\lambda$, $\sigma$ and $C_{overlap}$. For example, priors on $\lambda$ and $\sigma$ for correlating $\delta^{13}C$ curves could be meaningfully specified from pre-existing reconstructed $\delta^{13}C$ composite curves.*

5. Lines 228-232 indicate that uncertainty may not increase with distance from absolute age constraints. I agree with the previous reviewer that this seems to indicate a significant flaw in the methodology. It may again be due to the particular implementation – perhaps the density of spline knots should increase with distance away from the absolute age constraints (although it is not clear how quickly).

A similar issue arises close to sequence boundaries, around which time tends to be compressed in the stratigraphic record. Spline knots might be more densely distributed around such boundaries, but again it is not clear how dense they should be. This is another case where defining prior information is difficult, and requires more study (similar to the comment above).

We agree with both reviewers that the lack of increasing uncertainty away from the absolute age constraints is a flaw that needs to be addressed in future improvements to the StratoBayes methodology and software. We do not think that increasing (or decreasing) the number of knots away from age constraints or sequence boundaries could reliably solve this, especially since, as the reviewer points out, it is not clear how quickly the knot density should change.

We believe that our existing, slightly modified discussion of challenges related to the proxy and sedimentary record (section 5.2.3) sufficiently addresses the reviewer's comment:

*StratoBayes introduces a simplification in modelling sedimentary histories by forcing uniform sedimentation rates within pre-defined segments of a stratigraphic section. An effect of this simplification can be seen in the age-depth plots in Fig. 7: Due to sedimentation rates being modelled as uniform within stratigraphic partitions, the uncertainty of age estimates does not necessarily increase away from the radiometric dates. We acknowledge that this may underestimate the uncertainty associated with potential sedimentation rate variability (De Vleeschouwer and Parnell, 2014), especially when allowing for few sedimentation rate changes. Similarly, our method currently only allows for specifying potential hiatuses with an unknown duration at fixed, predetermined heights.*

*In principle, our method could be used to divide stratigraphic sections into an arbitrary number of segments with differing sedimentation rates, and with an arbitrary number of*

*potential hiatuses. In practice, estimating the parameters of a model with more than a low double-digit number of alignment parameters (shift parameters, sedimentation rates, hiatuses) represents a challenge for the current implementation of the MCMC algorithm within StratoBayes, as finding and exploring the posterior becomes increasingly difficult as more parameters are added. This limitation could be alleviated by incorporating MCMC methods suited for higher dimensional problems and difficult posterior geometries. Alternatively, a continuous process model such as the compound Poisson-gamma process of BChron (Haslett and Parnell, 2008) might be integrated with our model for the proxy signal, but again the complexity of the MCMC would increase. Another approach would be to divide the alignment problem into sub-problems, e.g. by multiple pairwise correlation of sites (e.g. Hagen et al., 2024; Sylvester, 2023), or by correlating shorter subsections.*

**Minor Comments:**

Line 130-131: This sentence needs some explanation; the main text should be understandable without having to read the Appendices.

We have expanded this sentence to make it clearer (lines 136 – 139), but still refer to the Appendices for a more thorough explanation, as a detailed description of the MCMC implementation will not be of interest for most readers:

*To ensure thorough exploration of the parameter space, we employ parallel tempering, i.e. we run multiple chains in parallel, flattening the likelihood of the tempered (hot) chains which can therefore move between different posterior modes, and frequently propose swaps between chains. For the posterior estimates, we retain samples only from the primary (cold) chain.*

Fig. 1 caption: as far as I can tell, both alpha and gamma are used in the main text before they are defined, other than in this figure caption.

We have corrected this oversight and added a definition for alpha and gamma where they first appear in the main text (lines 163 – 164):

*The knots for the spline can be distributed across the reference height range that the converted measurement heights occupy for a specific combination of shift parameters ($\alpha$) and scale parameters ($\gamma$, i.e. relative sedimentation rates).*

REFERENCES

C. E. Bond. Uncertainty in structural interpretation: Lessons to be learnt, Journal of Structural Geology, 74, 2015, https://doi.org/10.1016/j.jsg.2015.03.003

C. E. Bond, R.J. Lunn, Z.K. Shipton, A.D. Lunn; What makes an expert effective at interpreting seismic images?. Geology 2012;; 40 (1): 75–78. doi: https://doi.org/10.1130/G32375.1

A. Curtis, 2012. The science of subjectivity. Geology. 40, pp. 95-96. doi: 10.1130/focus012012.1

Andrew Curtis, Hugo Bloem, Rachel Wood, Fred Toby Bowyer, Graham Anthony Shields, Ying Zhou, Mariana Yilales, Daniel Tetzlaff, 2025. Natural sampling and aliasing of marine geochemical signals. Scientific Reports, 15:760, DOI: 10.1038/s41598-024-84871-6

A. Curtis and R. Wood 2004. Optimal elicitation of probabilistic information from experts. In, Geological Prior Information, A. Curtis and R. Wood ed's. Geol. Soc. Lond. Special Publication, Vol. 239; pp. 127-145; DOI 10.1144/GSL.SP.2004.239.01.09

D. Polson and A. Curtis, 2010. Dynamics of uncertainty in geological interpretation. Journal of the Geological Society, London, Vol. 167, pp. 5-10. doi: 10.1144/0016-76492009-055 M.

Walker and A. Curtis, 2014. Expert elicitation of geological spatial statistics using genetic algorithms. Geophys. J. Int., 198, pp.342–356, doi: 10.1093/gji/ggu132